# Identifying microbiota community patterns important for plant protection using synthetic communities and machine learning

Barbara Emmenegger[1,2], Julien Massoni[1,2] ✉, Christine M. Pestalozzi [1,2], Miriam Bortfeld-Miller[1], Benjamin A. Maier[1] & Julia A. Vorholt [1] ✉

Plant-associated microbiomes contribute to important ecosystem functions such as host resistance to biotic and abiotic stresses. The factors that determine such community outcomes are inherently difficult to identify under complex environmental conditions. In this study, we present an experimental and analytical approach to explore microbiota properties relevant for a microbiota-conferred host phenotype, here plant protection, in a reductionist system. We screened 136 randomly assembled synthetic communities (SynComs) of five bacterial strains each, followed by classification and regression analyses as well as empirical validation to test potential explanatory factors of community structure and composition, including evenness, total commensal colonization, phylogenetic diversity, and strain identity. We find strain identity to be the most important predictor of pathogen reduction, with machine learning algorithms improving performances compared to random classifications (94-100% versus 32% recall) and non-modelled predictions (0.79-1.06 versus 1.5 RMSE). Further experimental validation confirms three strains as the main drivers of pathogen reduction and two additional strains that confer protection in combination. Beyond the specific application presented in our study, we provide a framework that can be adapted to help determine features relevant for microbiota function in other biological systems.

Microorganisms occur in virtually all habitats, such as soil or air, and colonize multicellular organisms including plants, vertebrates, and invertebrates, assembling into communities termed microbiota[1,2]. These microbial communities are important for host development and health and more generally ecosystem functions[3–7]. The gut microbiota of healthy individuals, for example, confers protection against pathogens[8–11], also referred to as colonization resistance[12], aids in the training of the immune system[13–15], and is crucial for the digestion of food[16,17]. Similarly, plant-associated microbiota impact the state of their hosts by increasing nutrient availability[18,19], priming the plant immune system[20,21], alleviating biotic and abiotic stresses[22–24] - or impacting flowering time[25,26].

The realization that host-associated microbes affect host phenotypes led to the prospect that harnessing knowledge in this research area can help in biodiversity conservation[27,28] and engineering of microbiomes could be an effective and sustainable approach to address various challenges in medicine and ecosystem functions, including agriculture[29–34]. The identification of relevant microbiota properties for host traits is therefore a primary objective in microbiome research. Factors such as community composition and structure including species identity, richness, evenness as well as phylogenetic and functional diversities have been shown to play a role in emergent properties in various ecological systems[35–37]. However, it is inherently challenging to identify the primary drivers and they may

[1]Institute of Microbiology, ETH Zurich, Zurich, Switzerland. [2]These authors contributed equally: Barbara Emmenegger, Julien Massoni, Christine M. Pestalozzi. ✉e-mail: jmassoni@ethz.ch; jvorholt@ethz.ch

differ across biological systems[38]. In the case of microbiota-associated host phenotypes, community pattern or strain presence are often postulated to be crucial, based on observational studies that correlate the composition of microbial communities and host phenotypes[5,23,39,40]. But in order to establish causal relationships, experiments are required in which interacting partners can be manipulated[1,41,42]. As a first step in investigating the causality of host-microbiota interactions, it is common practice to add individual microbial strains to the host to assess the effects on the phenotype of interest, such as resistance to pathogen colonization. This has led to the identification of a suite of microbes that are able to protect hosts against specific pathogens. Evidently, examining microbes individually falls short in providing information about the effect of a microbe in the presence of varying microbiota members, thus warranting the need to examine the role of overall community properties[43–49].

Synthetic microbial community (SynCom) experiments offer the opportunity of testing hypotheses in reproducible conditions[1]. Such experiments benefit from reference collections that allow the assembly of synthetic communities relevant for microbiota-associated host phenotypes[36,46,50–52]. One such collection is the *Arabidopsis thaliana* leaf collection (*At*-LSPHERE) of environmentally representative bacterial strains isolated from healthy Arabidopsis plants[53] that has already been used to compose SynComs of varying composition and complexity[36,46,52–54].

In this study, we present an experimental and analytical approach to address the question of which microbiota features support plant protection across different biotic contexts. To do so, we randomly composed SynComs of the *At*-LSPHERE that we used to inoculate plants, and challenged them with the foliar pathogenic bacterium *Pseudomonas syringae* DC3000[20,22]. We investigated four microbiota features—absolute commensal colonization, community evenness, phylogenetic diversity and strain identity – to predict and validate plant protection outcomes (i.e., colonization resistance). We also provide a detailed rationale and calculations for the general design used in this study to facilitate its implementation in other host microbiota systems.

## Results
### Experimental design to screen random synthetic communities for plant protection and pilot experiment
To test whether microbial community characteristics relevant to a microbiota-conferred host phenotype can be identified using random synthetic communities and machine-learning analyses, we set up a screen for plant protection in a gnotobiotic model system (Fig. 1). The plant protection assays consisted in growing axenic *A. thaliana* Col-0[55], inoculation with randomly assembled synthetic communities (SynComs) of *At*-LSPHERE bacterial strains[53], and infection with a low dose of the pathogen *Pseudomonas syringae* DC3000 (Pst), as previously established[20,22,45] (Supplementary Fig. 1a, b) (see Methods for further details).

To set up the screen of randomly assembled synthetic communities, several parameters were considered in the experimental design, including the size of the pool of strains to assemble into SynComs (i.e., number of strains to draw from) and the richness of SynComs (i.e., number of community members), which will affect the frequency of strains in different SynComs (Fig. 2a).

A prerequisite for the statistical analyses was the observation of differences in pathogen colonization (our readout for plant protection) across SynCom treatments (Fig. 1b), which we expected to occur in small sized SynComs of 3–10 strains[45]. We opted to keep the size of the SynCom inoculum (i.e., the number of strains in a SynCom) constant to remove variation in SynCom complexity as a variable. In addition, we wanted to quantify the absolute abundances of the different SynCom members and the pathogen and opted for communities of five community members to be able to readily distinguish all strains in random SynComs based on colony morphologies (for more details see Methods).

A pilot experiment was conducted, in which we tested pathogen outcomes of 17 distinct and randomly assembled SynComs of five strains each (Mini5SynComs) taken from the *At*-LSPHERE collection (see Methods). Analysis of pathogen luminescence as a proxy for pathogen numbers showed variation in luminescence signals between the different Mini5SynComs (e.g., Welch's ANOVA at 12 days post infection (dpi), *p* value = $6.2 \times 10^{-7}$) (Supplementary Data 1, Supplementary Fig. 2). Additionally, 5 of the 17 Mini5SynComs did not show significantly elevated luminescence signals relative to the background signal of uninfected plants (one-tailed Welch's *t* tests, unadjusted *p* value > 0.05). Thus CFU enumeration rather than pathogen luminescence was more sensitive and provided a higher range of pathogen outcomes when comparing Mini5SynComs. Overall, the experiment indicated that Mini5SynComs are suitable to generate variation in pathogen colonization, a requisite for our screening approach.

Another important aspect of the experimental design was the replication scheme for each Mini5SynCom, which was also evaluated with the pilot experiment (see "Data analysis" in Methods). Each Mini5SynCom was inoculated onto 16 plants distributed among four microboxes to control for potential differences between microboxes. At 12 dpi, 14 out of 19 treatments had no significant box-to-box variation of pathogen luminescence (*p* value > 0.05) (Supplementary Fig. 3). Therefore, one microbox was used per Mini5SynCom in the main experimental setup to increase the total number of alternative Mini5SynComs screened.

The size of the strain pool (i.e., the strains drawn from to assemble in random SynComs), was another important component of our experimental design that determined the prevalence of each strain across Mini5SynComs. To reach the presence of strains in about 20 independent Mini5SynComs, the calculated strain pool size was 35 or lower for a total of 136 communities, which we considered experimentally feasible (Fig. 2a). The selected strains spanned the phylogenetic diversity of the *At*-LSPHERE[53] (Fig. 2b, Supplementary Fig. 1c,

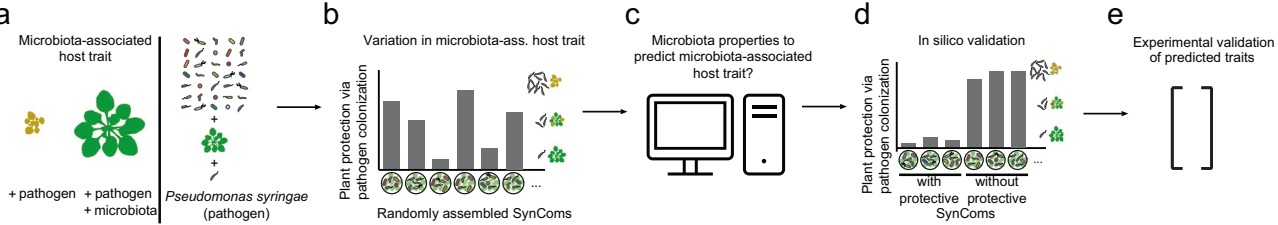

**Fig. 1 | Overview of screening strategy.** The screening and analysis approach consists in (**a**) an experimental system of *A. thaliana* as host, a pathogen (*Pseudomonas syringae*) and a bacterial strain collection; (**b**) randomly constituted synthetic communities with which the host is inoculated and measurements of the community and host traits of interest; (**c**) statistical modeling to identify properties of communities which correlate with variation in the trait of interest (e.g., machine learning); (**d**) validation of models with an independent dataset; and (**e**) empirical validation of the properties of the communities identified as potentially impacting the host trait, here pathogen colonization.

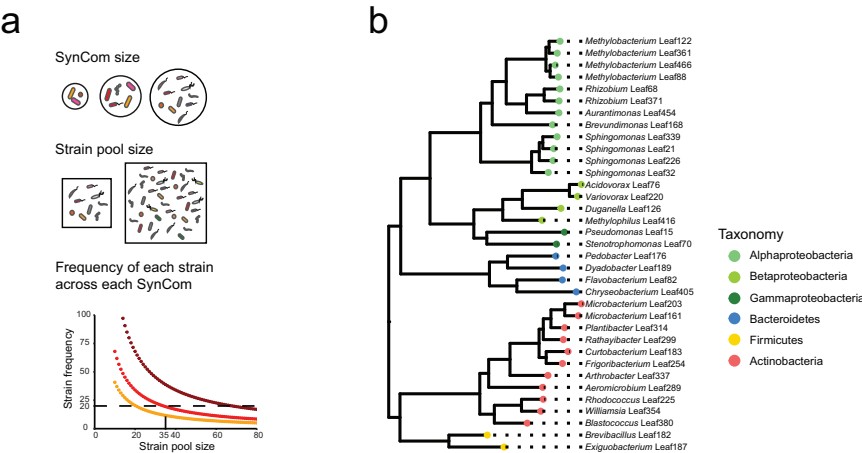

**Fig. 2 | Experimental constraints and strain selection. a** SynCom and strain-pool sizes are the two parameters affecting the expected prevalence of strains in SynComs for a known number of experimental units. **b** Phylogenetic diversity of the collection of strains (SynCom-35) used for the screening experiment.

## Screening of random Mini5SynComs for plant protection

The screen of 136 randomly assembled Mini5SynComs was partitioned into two independent experiments (experiments 1 and 2). For each plant, fresh weight and bacterial colonization of both pathogen and commensals (i.e., Mini5SynCom strains) were determined at 14 dpi (Fig. 3, Source Data). In each experiment, controls included axenic non-infected (axenic NI), axenic infected (axenic), a SynCom of the entire pool of 35 strains (SynCom-35), as well as SynCom-Low and SynCom-High of the pilot experiment (see "Synthetic community assembly and controls of the Mini5SynCom screen" in Methods for details).

Median pathogen colonization (median of plant measurements per treatments) ranged from four to nine orders of magnitude in CFU g$^{-1}$ plant fresh weight across all SynComs tested (Fig. 3a). The controls SynCom-Low and SynCom-35 strongly reduced pathogen colonization, being in the first quartile of communities ranked according to the median pathogen colonization, while axenic infected controls had the highest median pathogen colonization of their respective experiments. In contrast to pathogen colonization, overall commensal colonization was more similar between different communities and ranged from seven to nine orders of magnitude in CFU g$^{-1}$ plant fresh weight (Fig. 3b). The median plant weight was between 13.0 and 68.1 mg (Fig. 3c; Source Data), with axenic-infected plants showing the lowest median plant weight (18.4 and 13.0 mg for experiment 1 and 2, respectively).

## Correlation analysis of overall commensal colonization, community evenness, or phylogenetic diversity to pathogen abundance

Pathogen colonization was investigated for its association with overall Mini5SynCom characteristics, specifically overall commensal colonization, community evenness and phylogenetic diversity. Linear mixed models were fitted with the log$_{10}$-transformed pathogen colonization as dependent variable, and either the log$_{10}$-transformed commensal colonization (centered), evenness or phylogenetic diversity as fixed effects (see "Data analysis" in Methods for details). For overall commensal colonization, the four best models (delta AIC < 4) supported a significant increase in commensal colonization of one order of magnitude with an equivalent increase in pathogen colonization considering all analyzed Mini5SynComs (Fig. 4a; Supplementary Data 3). The standard errors of the corresponding coefficients were consistently below one order of magnitude, and the standard deviations of residuals were one order of magnitude (Supplementary Data 3, Sup-

plementary Fig. 4a). Therefore, a positive relationship can be concluded between pathogen and commensal colonization, which is consistent over different community compositions (in the best model: commensal effect = 1.4, standard deviation of commensal effect = 0.6; Supplementary Data 3).

Variation in pathogen colonization was also associated with changes in evenness of Mini5SynComs. However, the three best models (delta AIC < 4) supported that the entire range of the evenness function (zero to one) corresponded within one order of magnitude to the decrease in pathogen colonization, which is a small value compared to the six orders of magnitudes spanned by pathogen colonization across Mini5SynComs (p values < 0.01; Fig. 4b; Supplementary Data 3). In addition, when including only communities with all strains being present above level of detection and no ambiguous counts, the relationship between evenness and pathogen colonization was not significant (p values > 0.05; Supplementary Data 3). For all models fitted to the entire dataset, the standard errors of the corresponding coefficients were consistently less than one order of magnitude, and the standard deviations of residuals were one order of magnitude (Supplementary Data 3, Supplementary Fig. 4b). The relationship between evenness and pathogen colonization was less consistent across different community compositions with a relatively high estimate of the standard deviation of this fixed effect (in the best model: evenness effect = -1.3, standard deviation of the evenness effect = 2.2; Supplementary Data 3).

In contrast to commensal colonization and evenness, we did not observe significant variation in pathogen colonization with phylogenetic diversity in all best models, except for one of the four best models with the weighted mean pairwise distance (mpd) as fixed effect and all communities without ambiguous counts (delta AIC < 4; Supplementary Data 3, Fig. 4c).

## High predictiveness of pathogen reduction by synthetic microbiota composition using machine learning

Apart from the general community parameters, we tested whether community composition (i.e., strain identity) was associated with pathogen colonization. We applied supervised machine-learning algorithms, i.e. random forest (RF) and elastic-net regularized generalized linear models (GLMNet), to train models to predict the outcome of pathogen infection based on Mini5SynCom composition (see "Data analysis" in Methods for details).

The distribution of pathogen colonization was bimodal within experiments 1 and 2 (Fig. 5a, b) suggesting two biological groups with low and high pathogen colonization. To test the biological relevance of

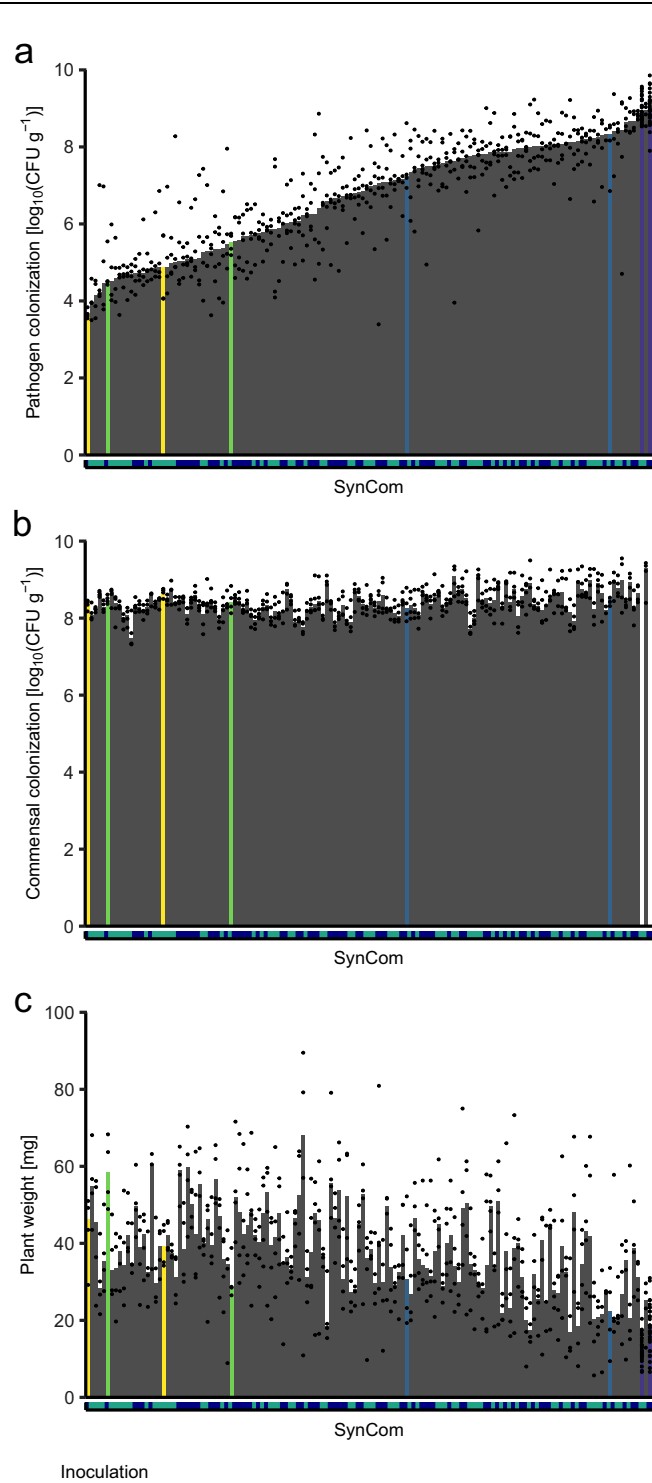

**Fig. 3 | Results of screen of 136 randomly assembled Mini5SynComs. a** Pathogen colonization, (**b**) commensal colonization, (**c**) plant weight. Bars are medians with individual plant measurements superimposed as a scatter plot; x-axes represent individual treatments colored according to experimental rounds. The order of treatments is fixed across all panels and ordered in ascending pathogen colonization. Exp experiment.

these bimodal distributions (i.e., observation not by chance), we bootstrapped the data 1000 times and confirmed global minima of interest in virtually all replicates (Supplementary Data 4). We thus classified the samples according to the global minimum of pathogen colonization in each distribution into "protected" (pathogen colonization lower than the minimum, positive class) and "non-protected" (pathogen colonization equal to or higher than the minimum, negative class) to train classifiers (Fig. 5a, b; Supplementary Data 5). In addition, we trained regression algorithms to predict the normalized pathogen reduction of a Mini5SynCom (see Methods). As predictors (features), we used presence/absence or absolute abundances of Mini5SynCom members ("colonization"). The algorithms were either trained using measurements of individual plants or the calculated median values derived from four plants of a box (i.e., Mini5SynCom treatment). All 12 model-algorithm combinations (Supplementary Fig. 5) were trained using 10 rounds of 5-fold cross validation with repetition of eight different seeds (see "Data analysis" in Methods for details).

Subsequently, we evaluated the performance of the trained models by generating an independent test dataset (Experiment 3; see "Data analysis" in Methods), a new set of 70 random Mini5SynComs, none of which had a composition present in the training sets. The test dataset had similar characteristics as the training dataset and was not seen by the algorithms during training (Supplementary Fig. 6, Supplementary Data 4, 5, Source Data). Classification using presence/absence or commensal colonization led to substantially better predictions than random classification of samples ("No Model"), with 84-93% of samples correctly classified as either "protective" or "not protective" (Fig. 5c, d, Supplementary Data 6) compared to 51 or 56% by random classifiers. The fraction of true protective samples in the set of samples predicted as protective (i.e., precision) ranged from 72 to 82%, while random classification showed a precision of 42% and 35% on median and individual values, respectively. The fraction of correctly predicted protective samples in the set of true protective samples (i.e., recall) was 94% or 100%, with only one recall at 73%. In comparison, the recalls of the random classifiers were 32%. The fraction of correctly predicted non-protective Mini5SynComs (i.e., specificity) was consistently above 84%, versus less than 69% by random classifications. Regression analyses were also better performing than predictions based on the global average of pathogen colonization, (i.e., average calculated from the plant or median mix data, "No Model") with a root-mean-squared error (RMSE) ranging from 0.79 to 1.06 for the trained analyses compared to an RMSE of 1.5 for the global average. This translates to an error of one versus two orders of magnitude of trained versus untrained models, respectively (Fig. 5e, f; Supplementary Data 7). Regression analyses using absolute colonization of commensal strains as predictors showed no improvement over predictions using presence/absence of strains. This suggests that in our system, the presence of strains are predictive of pathogen colonization outcomes.

### Three strains found to be the most important in machine learning algorithms strongly reduce pathogen colonization

After evaluating the machine learning algorithms (Fig. 5c–f), we wanted to specifically validate the strains found to be the most important for the predictions in a targeted manner (Fig. 1d). All the 12 analyses converged to support that three strains, i.e., *Acidovorax* Leaf76, *Rhizobium* Leaf68, and *Pseudomonas* Leaf15, were the most important features to predict pathogen colonization or protection class (Fig. 6a, b; Supplementary Fig. 7; Supplementary Data 8). Only one seed of the GLMNet regression analyses using absolute commensal colonization of individual plants had a divergent result (Supplementary Fig. 7).

We empirically tested the ability of the three strains *Acidovorax* Leaf76, *Rhizobium* Leaf68, and *Pseudomonas* Leaf15 to reduce pathogen colonization. We also investigated the strain *Rhizobium* Leaf371 which ranked fourth in relative feature importance of regression analyses. We validated the strains individually by testing whether they

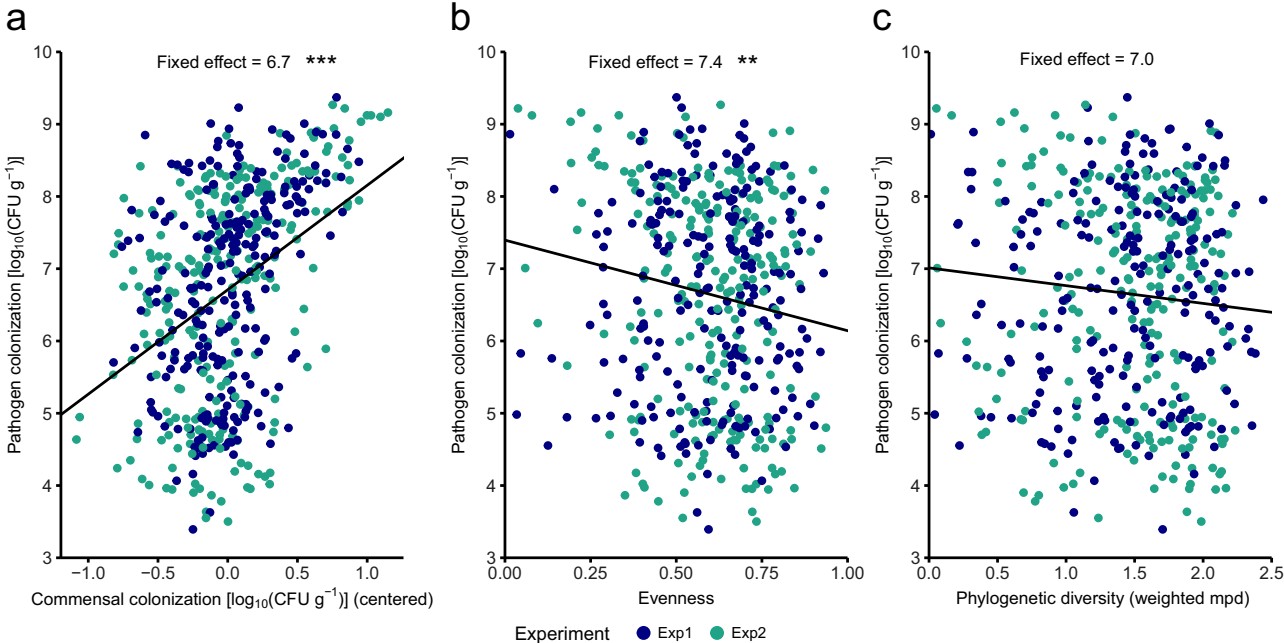

**Fig. 4 | Correlation among pathogen colonization, overall commensal colonization, evenness, and phylogenetic diversity.** Regression lines were obtained from best mixed effect models (see Methods for more details). **a** Pathogen colonization against overall Mini5SynCom commensal colonization. **b** Pathogen colonization against Mini5SynCom evenness. **c** Pathogen colonization against phylogenetic diversity (weighted mean pairwise distances) weighted with strain abundances. Significance levels of the coefficient of the fixed effect, obtained with two-sided $t$ tests, are indicated above plots: 0 '***' 0.001 '**' 0.01 '*' 0.05 '.'. Exp experiment, mpd mean pairwise distance.

could significantly reduce pathogen colonization (see "Validation experiment of machine learning results" in Methods). We also tested their synergic effect in binary combinations as well as all four strains together within the same SynCom. To demonstrate that the identified strains provided better protection than other strains, alone or in combination, we assessed the pathogen reduction potential of four randomly selected strains in parallel.

Consistent with the relative feature importance obtained by the machine learning algorithms, the three best-predictive strains *Acidovorax* Leaf76, *Rhizobium* Leaf68, *Pseudomonas* Leaf15 significantly reduced pathogen colonization by two orders of magnitude compared with axenic controls ($p$ value < $1.6 \times 10^{-17}$ after Bonferroni correction in the single best model with delta AIC < 4; Fig. 7a; Supplementary Data 10, Supplementary Data 11). We therefore termed these three strains "pathogen-reducing strains" (PR strains). *Rhizobium* Leaf371 that had lower relative feature importance in the machine learning, also significantly reduced pathogen colonization; however, by one order of magnitude (Bonferroni-adjusted $p$ value = $7.3 \times 10^{-6}$), suggesting an intermediate pathogen reduction level. In contrast, only one of four randomly selected strains significantly reduced pathogen colonization (Bonferroni-adjusted $p$ value = 0.05) by less than one order of magnitude, confirming that the strains identified by machine learning reduce pathogen colonization by a higher level than the random strains tested (Fig. 7b; Supplementary Data 12).

Furthermore, *Rhizobium* Leaf68 and *Acidovorax* Leaf76 significantly reduced pathogen colonization by one order of magnitude when applied in combination compared to their individual treatments ($p$ value < 0.005 after Bonferroni correction; Fig. 7a, Supplementary Data 11). Notably, this combination as well as the SynCom of the three PR strains in combination with *Rhizobium* Leaf371 reduced pathogen colonization by about four orders of magnitude, which was not significantly different from the pathogen reduction of the SynCom-35 (Fig. 7a, Supplementary Data 11). Other binary PR-strain combinations showed non-significant lower pathogen colonization compared to their individual treatments despite a significant reduction of pathogen

colonization compared to axenic controls (Fig. 7a, Supplementary Data 11). Five out of the seven random-strain combinations significantly reduced pathogen colonization compared to the axenic controls, however to a maximum of one order of magnitude observed for the four-strain combination (Fig. 7b, Supplementary Data 12).

Overall, these experiments confirmed the validity of the machine learning approach to identify important strains and revealed synergic effects of individual PR strains.

### Refined data analysis and experimental validation reveal combination of strains reducing pathogen colonization

After experimental validation of the top featuring strains in pathogen reduction, we reanalyzed the data from the original screen in the light of machine learning and experimental validation results (Fig. 1e). For this, we split the median pathogen colonization data of the screen experiments 1 and 2 into two groups. The group "PR Strains" was composed of Mini5SynComs containing any PR strains (*Pseudomonas* Leaf15, *Rhizobium* Leaf68, and *Acidovorax* Leaf76)— individually or in combinations—while the group "Others" contained the remaining Mini5SynComs. We observed a clear separation of treatments with lower pathogen colonization, reminiscent of the bimodal pathogen distribution (Fig. 5a, b), with the "PR Strains" showing a median of $1.9 \times 10^5$ CFU g$^{-1}$ plant fresh weight, and a higher pathogen colonization level of the "Others" group (median of $6.2 \times 10^7$ CFU g$^{-1}$ plant fresh weight) (Fig. 8a). Interestingly, the "Others" group had a skewed distribution with four Mini5SynComs in the low pathogen colonization region, suggesting additional features not yet identified with the applied machine learning approaches (red circle in Fig. 8a). Analysis of the composition of the four circled Mini5SynComs revealed that all contained *Rhizobium* Leaf371 and three out of four additionally contained *Arthrobacter* Leaf337 (Fig. 8b). *Arthrobacter* Leaf337 was found to be the fifth most important strain in the machine learning (Fig. 6a, b) with a slightly lower average relative feature importance (18%) compared to that of *Rhizobium* Leaf371 (21%). From this observation, we visualized the

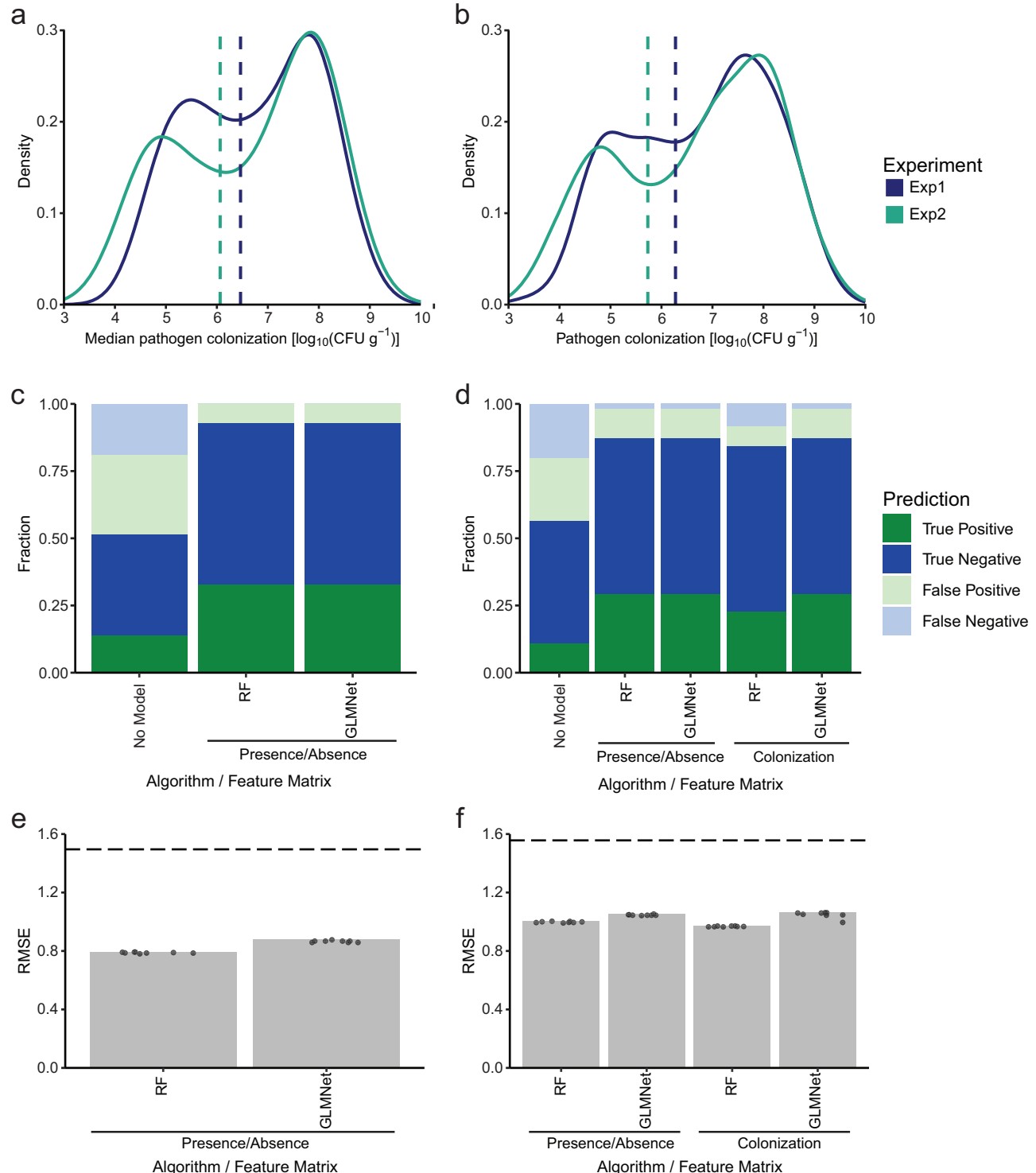

**Fig. 5 | Predictive outcome of machine learning. a, b** Density curves of pathogen colonization with global minima used for splitting into "protected" (positive) and "non-protected" (negative) classes presented as vertical dashed lines.
**c, d** Performances of classification algorithms compared to a random classification (i.e., "No Model") based on presence/absence of Mini5SynCom members.
**e, f** Root mean square errors (RMSE) of the regression algorithms with dashed lines corresponding to predictions based on the global average of pathogen colonization ("No Model") based on presence/absence of Mini5SynCom members or absolute abundance of Mini5SynCom members ("colonization").
**a, c, e** Results derived from algorithms trained on the median of pathogen colonization for each treatment. **b, d, f** Results derived from algorithms trained on pathogen colonization of individual plants. RF random forest, GMLNet elastic net regularized generalized linear model.

pathogen colonization of plants inoculated with Mini5SynComs containing either *Arthrobacter* Leaf337, *Rhizobium* Leaf371 or their combination (Fig. 8c). The communities containing both strains tended to show a lower pathogen colonization than communities containing either strain alone, suggesting a potential additive effect of *Rhizobium* Leaf371 and *Arthrobacter* Leaf337.

The hypothesis that *Rhizobium* Leaf371 and *Arthrobacter* Leaf337 together significantly affect pathogen colonization was experimentally

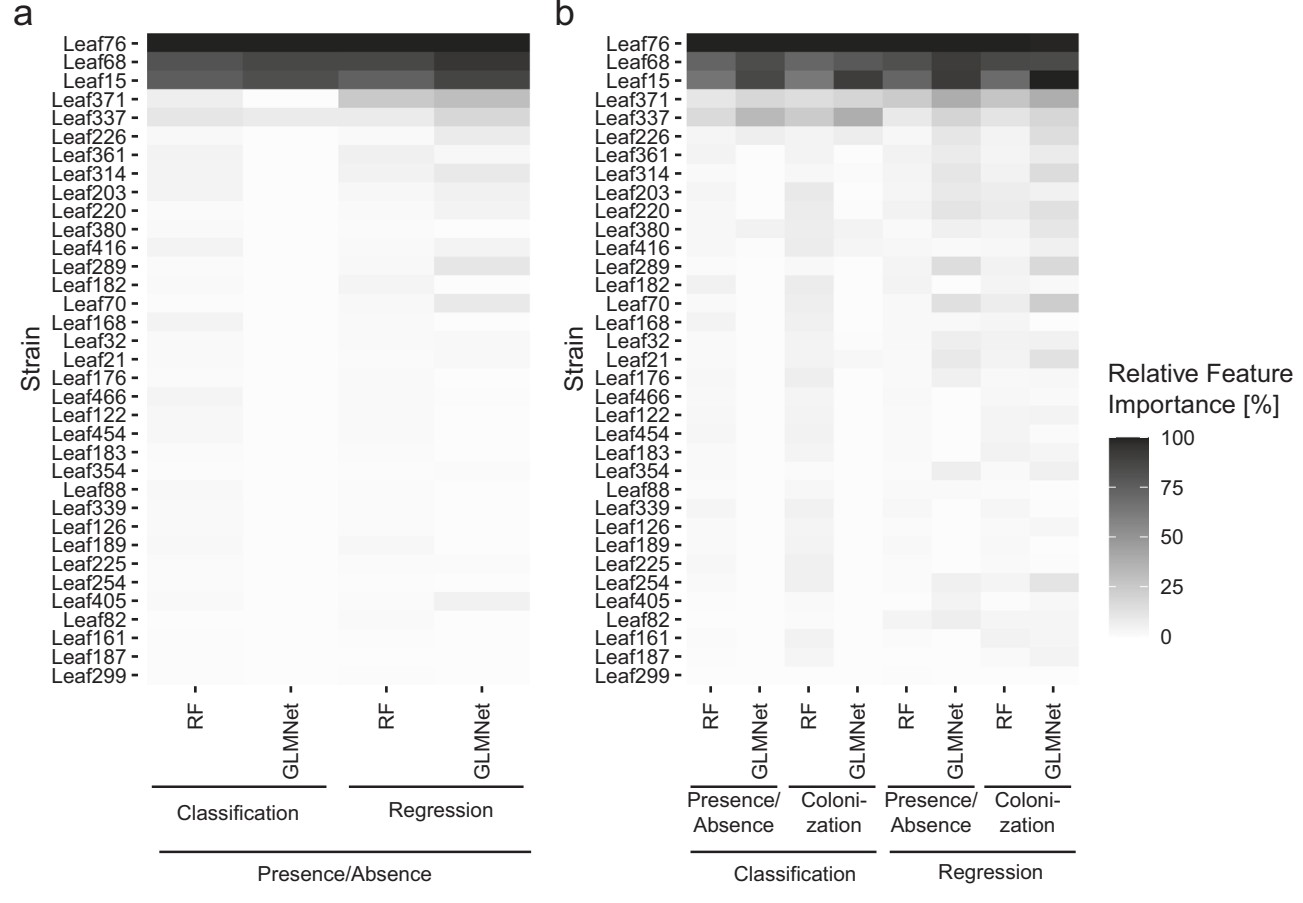

**Fig. 6 | Relative feature importances of trained machine learning algorithms.** Medians of the relative feature importances calculated across eight seeds per algorithm; strains are ordered according to their median relative importance across all analyses. **a** Relative feature importances derived from algorithms trained on the median of pathogen colonization for each treatment. **b** Relative feature importances derived from algorithms trained on pathogen colonization of individual plants. RF random forest, GMLNet elastic net regularized generalized linear model.

validated. The two strains were examined individually, in binary combination, as well as in three Mini5SynComs containing both strains (see "Validation experiment of synergic effect of strains" in Methods). The best two models with no random effect and a random intercept for the experiment effect (delta AIC < 4; see Methods; Supplementary Data 13) converged regarding the significance of coefficients (Supplementary Data 14). The combination of *Rhizobium* Leaf371 and *Arthrobacter* Leaf337 together, and in combination with other non-PR strains in the validation mixes 1 and 2 (ValMix 1, 2) significantly improved the pathogen reduction by one order of magnitude compared with treatments of individual strains ($p$ values ≤ 0.02 after Bonferroni correction; Fig. 8d; Supplementary Data 15). Mix6, reproduced from the original screen, induced a significant pathogen reduction compared to *Rhizobium* Leaf371 alone ($p$ value = 0.02 after Bonferroni correction), but not to *Arthrobacter* Leaf337 alone. Overall, the data demonstrate the value of pattern analysis after identifying individual strains through machine learning.

## Discussion

The complexity of interactions that take place in host-associated microbiota makes it difficult to identify the principle drivers of phenotypes of interest[56,57]. Therefore, the identification of microbiota patterns with desired impact on ecosystem functions, such as host phenotypes, remains a challenging task. Here, we explored the identification of features relevant for host phenotypes by starting from a community context which does not require a priori knowledge or

hypotheses on the mechanisms underlying microbiota-conferred traits of the host[58]. We showed that the screening of randomly composed SynComs can be used in combination with statistical modeling to identify community compositions with desired effects on host traits (here pathogen reduction; Fig. 1a).

The analysis of general community structure properties showed that an increase of commensal colonization positively correlated with an increase in pathogen colonization. This result is in agreement with previous observations and might be explained by the release of plant nutrients due to pathogen colonization which in turn promote commensal growth[59]. Both initial and realized phylogenetic diversity did not explain pathogen colonization. This absence of correlation might be due to the lack of strong phylogenetic signals in relevant functional traits or biotic interactions[60,61], or a disproportionate role of individual strains in plant protection as supported by the validation experiments[38,62] (Fig. 7a).

Our machine learning analyses of community composition identified strain identity as an important factor to predict protection outcomes and confirmed the robustness of strain effects to biotic variation (Fig. 6). We identified and validated three strains (PR strains), *Pseudomonas* Leaf15, *Rhizobium* Leaf68, *Acidovorax* Leaf76, that decreased pathogen colonization when present in communities in the phyllosphere of *A. thaliana* (Fig. 7), one of which, *Acidovorax* Leaf76, was not identified before, albeit using different experimental conditions[45]. Previous research has successfully utilized machine learning algorithms to identify individual microbial taxa associated

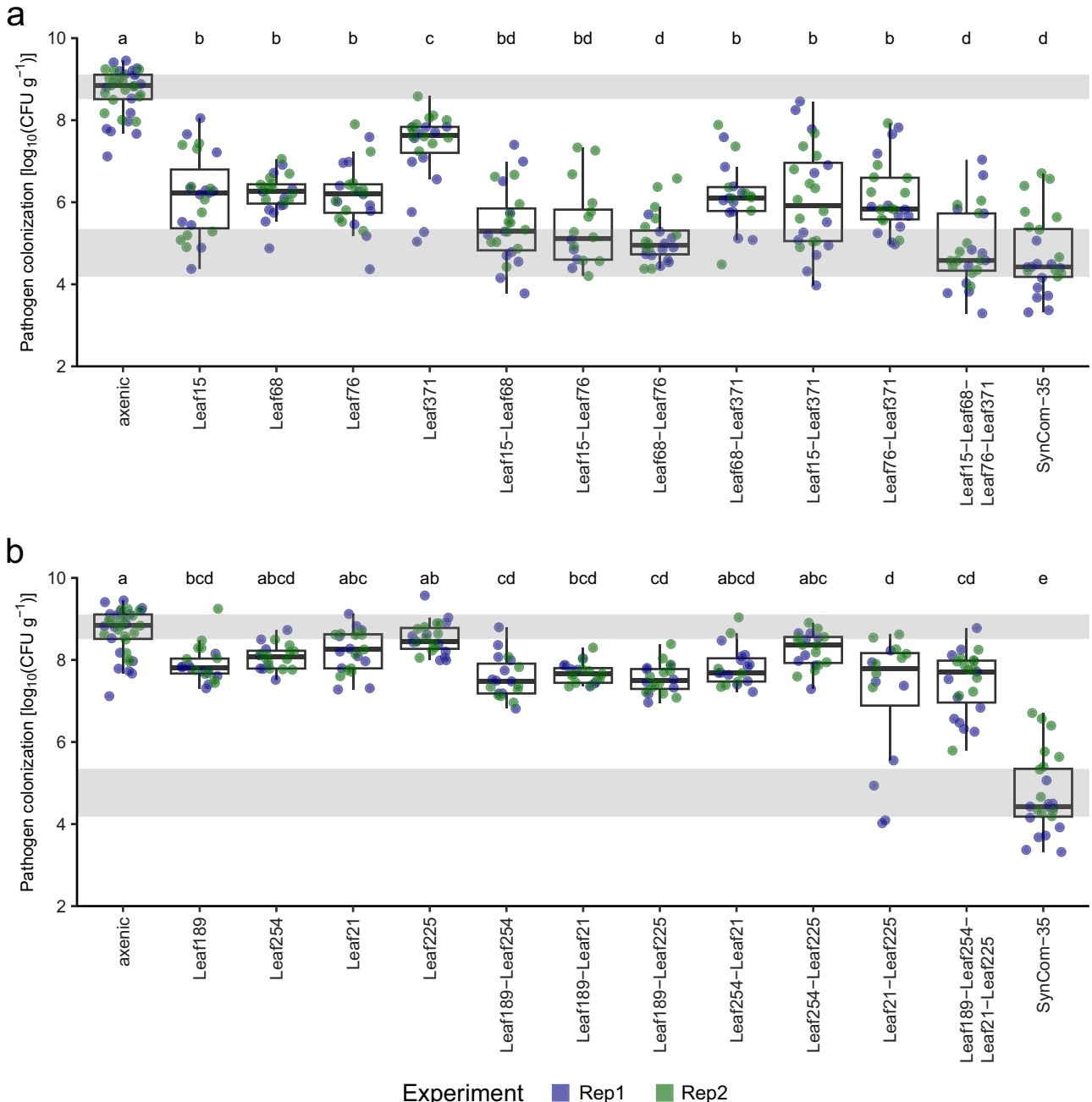

**Fig. 7 | Empirical validation of machine learning results. a** Boxplot of the pathogen colonization for each PR-strain treatment and controls (*n* = 24). **b** Boxplot of the pathogen colonization for random-strain treatments and controls (*n* = 24). The boxplots show median with hinges corresponding to first and third quartiles, and whiskers extending to 1.5 times the inter quartile range. The inter quartile ranges of the axenic and SynCom-35 controls are shaded in gray. Significant differences in pathogen colonization were estimated with the best model including strain inoculation as fixed effect, a random intercept for the experimental effect, and two-sided *t*-test procedures. Lettering corresponds to significance groups at a 0.05 level after Bonferroni correction with the whole family of pairwise comparisons in each panel. The axenic and SynCom-35 controls shown in both panels are duplicated data.

with specific properties, including ecosystem functions, disease-suppressive soil, plant organs, and agricultural management[7,58,63–66]. In this study we extrapolate this approach to host phenotype and synthetic communities. Random-forest and GLMNet approaches offer large flexibility in their respective statistical frameworks as they can treat the response variable as categorical, discrete, or continuous. Continuous and discrete treatments avoid the need for a priory definition of classes, which, on the contrary to the present case, can often be difficult to justify in the absence of a bimodal distribution. When data were split among two classes, "protected" and "non-protected", a simpler analytical approach might have consisted of identifying strains

being more frequent in the "protected" class with contingency tables (*e.g.*, chi2 or Fisher tests). However, because these approaches do not implement procedures against overfitting issues, machine learning algorithms should outperform these regarding the proportion of identified candidates as protective in subsequent validation experiments. Furthermore, estimating feature importance (*i.e.*, identifying strains crucial for predicting protection outcomes) in random forest analyses enables the detection of key strains for plant protection across all possible patterns of differential frequencies within protection classes. Although, the present screen was not designed to detect plant protection by strain combinations within communities,

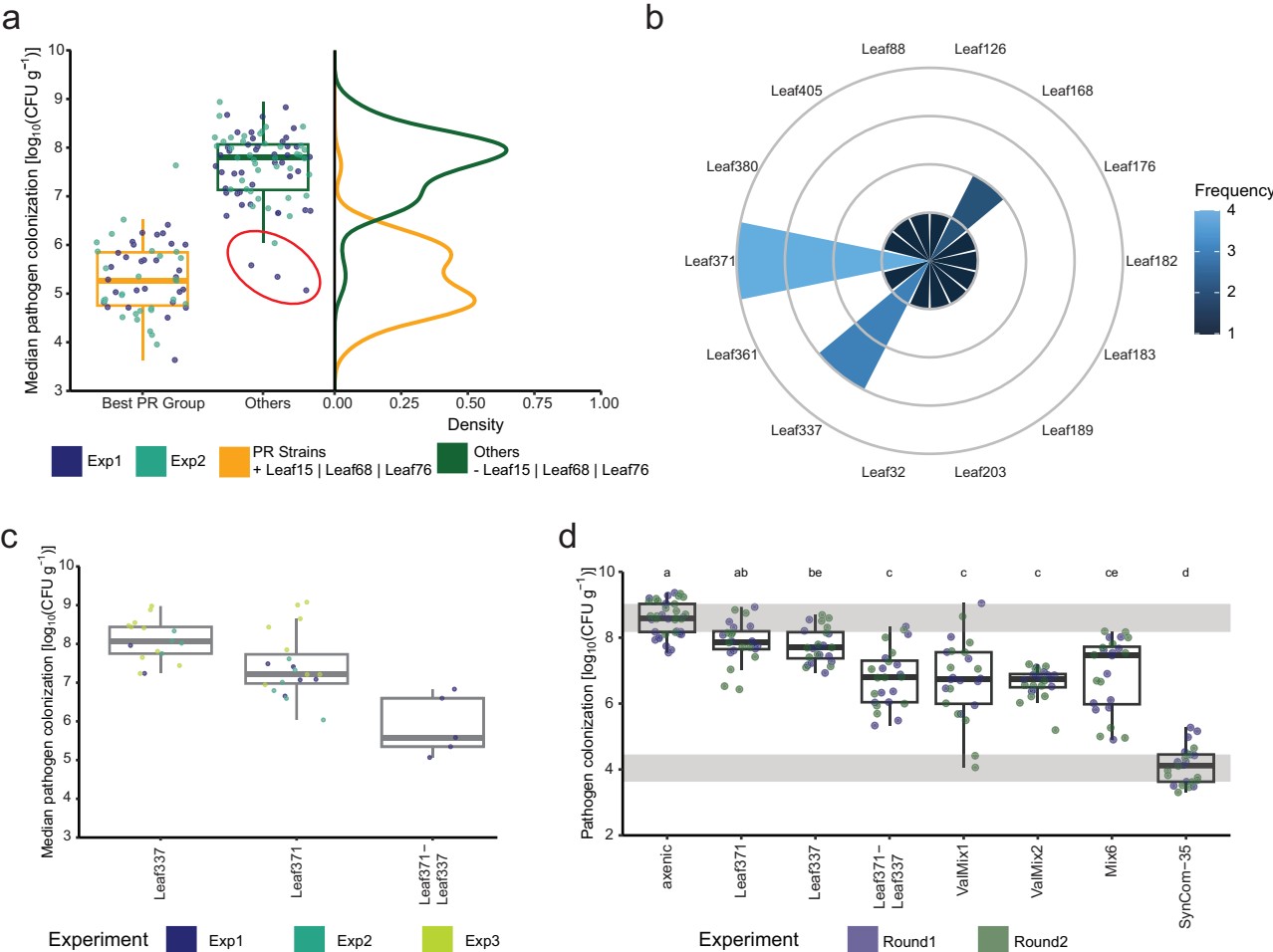

**Fig. 8 | Identification of additional two strain combination that reduces pathogen colonization. a** Comparison of the group of treatments that contained at least one of the best-pathogen-reducing strains, Leaf15, Leaf68, or Leaf76 ("PR Strains", $n = 57$), and treatments that did not include those strains ("Others", $n = 84$); The circled Mini5SynComs are those which are included in the tail of the distribution of the "Others" group. Shown are boxplots and density curves of pathogen colonization with each point corresponding to the median of one treatment box. **b** Frequency of strains present in the Mini5SynComs of the tail of the "Others" group distribution and circled in (**b**). **c** Boxplot of the median pathogen colonization with communities containing Leaf371 ($n = 19$), Leaf337 ($n = 17$), or their combination in the screen experiments 1 and 2 and test set (Exp3)

($n = 5$). **d** Boxplot of the pathogen colonization with individual strain inoculations, binary combination and new or repeated Mini5SynComs including Leaf371 and Leaf337 with random non-PR strains (ValMix 1,2, Mix6) (for more details see Methods) ($n = 24$). The interquartile ranges of the axenic and SynCom-35 controls are shaded in gray. Significant differences in pathogen colonization were estimated with the best model including strain inoculation as fixed effect, no random effect, and two-sided $t$ test procedures. Lettering corresponds to significance groups at a 0.05 level after Bonferroni correction with the whole family of pairwise comparisons in this panel. The boxplots show the median with hinges corresponding to first and third quartiles, and whiskers extending to 1.5 times the inter quartile range. Exp experiment, ValMix validation mix.

increasing the prevalence of specific combinations in the screen would allow investigations of their impact on plant protection using the same machine learning approaches. Strain interactions can be incorporated in GLMNet models, and combinations of strains can be features in Random Forest analyses. In addition, close investigation of specific permutation schemes might help to detect important combinations of features without explicit coding. For instance, after detection of a strain being important for accuracy, one could try to detect "helpers" and "cancellers" of the effect by measuring accuracy variations with value randomization (presence/absence, abundance) of other strains being present in the same SynComs. Furthermore, specific screens with all communities containing a specific PR strain and random companion strains could be designed to identify those synergic or antagonist effects from a large pool of bacteria applying the same machine learning technics. In this study, we nonetheless uncovered synergic effects of the PR strains (Fig. 7a) through data exploration and experimental validation. In addition, we found that *Rhizobium* Leaf371 and *Arthrobacter* Leaf337 had a significant protective effect on pathogen reduction when present together (Fig. 8).

The additive effects of the PR strains suggests that these may have complementary mechanisms to reduce pathogen colonization, which is an important finding for future SynCom design and application studies. Regarding mechanistic explanations, one way to limit pathogen colonization is for the protective strains to act through the plant by activating the plant defences[67–69]. Protective bacteria can also reduce pathogen invasion by resource competition[70–72], or by direct inhibition of the pathogen[23,73,74]. All three PR strains were previously shown to activate plant responses[75], contain the type VI secretion system, and *Rhizobium* Leaf68 and *Pseudomonas* Leaf15 were found to protect pattern-triggered-immunity plant mutants[45], which might indicate a contribution of microbe-microbe interactions in pathogen reduction or nutrient competition in the phyllosphere[22,54]. In addition, the synergic effect of the PR strains on pathogen reduction could also be a result of growth facilitation[52,54,55,76]. Further experiments will be needed to characterize the mechanisms behind plant protection identified in the present study. In addition, it will be interesting to investigate in how far the PR strains are applicable to other plant species.

Finally, because virtually all communities with PR strains had a reduced pathogen colonization suggesting the absence of canceling effects by other strains (Fig. 8a), we expect protection by PR strains to be robust and propagate to richer communities in *A. thaliana*. Consequently, our study provides the basis to design more complex protective synthetic communities and investigate potential emergent properties in richer microbiota. For instance, larger communities will allow more frequent selective and complementary processes hypothesized to sustain different aspects of the link between biodiversity and ecosystem functions[77,78].

In conclusion, strain identities and their combinations are important predictors for pathogen colonization in our system. The robustness of the results predicted by machine learning and the demonstration of causality through empirical validation indicates that the screening method is suitable and allows to identify beneficial strains or strain consortia, or potentially also features that only manifest at the community level. The knowledge may then be used to design SynComs for host applications and investigate the underlying mechanistic basis. Due to the simplicity and flexibility of the experimental and analytical design, the microbiota screening approach coupled with machine learning is applicable to host-microbiota systems more broadly, beyond the one presented in the biological example of the present study.

## Methods
### Plant growth conditions
In all experiments of the present study, *Arabidopsis thaliana* Col-0 were grown gnotobiotically as described previously[55,79]. Briefly, 140 ml heat-sterilized calcined clay (Diamond Pro Calcined Clay Drying Agent) was mixed with 60 ml filter-sterilized 0.5× Murashige and Skoog (MS) medium including vitamins, pH 7 (Duchefa, Cat.no. M0222.0050) in gamma-irradiated microboxes (Saco2, Cat.no. O118/80 + OD118 with XXL + (green) filter lid). Surface sterilized seeds were stratified at 4 °C for 4 d and 2–3 plants seeded at four spots. Plants were placed in growth chambers (CU-41L4, Percival) set to 22 °C and 54% relative humidity with a 11 h photoperiod. Light intensities were set to 180-200 $\mu$mol m$^{-2}$ s$^{-1}$ (400–700 nm, PAR; Philips Master TL-D 18 W/950 Graphica) and 5–6 $\mu$mol m$^{-2}$ s$^{-1}$ (280-400 nm, UV light; Sylvania Reptistar F18W/6500 K). Surplus seedlings were removed from each planted spot to have exactly four plants per box prior to inoculation (10 d). Plants were watered with 500 $\mu$l 0.5x MS on days 4, 17, 24 and 31, and harvested on day 38 (5.5 weeks old).

### Inoculation of plant with bacterial suspensions
Throughout this manuscript, the term "inoculation" is used to refer to treatment with commensal strains of the *At*-LSPHERE bacterial collection[53], isolated from healthy-looking environmental *A. thaliana* plants, whereas the term "infection" refers to spraying with the foliar pathogen *P. syringae*. Bacterial strains were streaked out on R2A agar (Sigma-Aldrich, Cat.no. 1004160500) supplemented with 0.5% (v/v) methanol (Sigma-Aldrich, Cat.no. 179337) and incubated at 22 °C for 6 days. Strains were resuspended individually in 1.5 ml 10 mM MgCl$_2$ (Sigma-Aldrich, Cat.no. 63068), and vortexed for 10 min. If required, suspensions were filtered through a sterile 10 $\mu$m filter (CellTrics, Sysmex Suisse AG, Cat.no. 04-004-2324) to remove aggregates. Suspensions were adjusted to OD$_{600}$ of 0.2 and combined in equal ratio to prepare the Mini5SynComs. The mixed Mini5SynCom suspensions were diluted 1:10 and used for inoculation. Axenic seedlings were mock-inoculated by pipetting 500 $\mu$l of bacterial suspension or 10 mM MgCl$_2$ onto the plants at 10 days post germination. To control the OD adjusted suspension and the inoculum, ten-fold dilution series were prepared, and spotted onto R2A agar supplemented with 0.5% methanol to determine colony-forming units (CFU).

### Plant infection
Infection inoculum of *Pseudomonas syringae* pv. *tomato* DC3000 *luxCDABE* (Pst)[80] was prepared as described by Innerebner et al. [22]. Briefly, a lawn of Pst was grown on King's B agar[81] at 28 °C overnight, resuspended in 10 ml 10 mM MgCl$_2$ and OD$_{600}$ adjusted to 0.001. The plants were sprayed at day 24 with either buffer (mock-infected, non-infected controls, NI) or with Pst suspension using a thin-layer chromatography reagent sprayer (Faust Laborbedarf AG). Each box was sprayed 6 times, which corresponded to roughly 10$^4$ pathogen CFUs per plant. Pathogen titer was assessed by CFU determination on King's B agar.

### Bacterial plant colonization by colony-forming units
Except for the pilot experiment (see section "Data analysis"), plants were harvested at 14 dpi (days post infection, corresponding to 38 days old plants). Plant weight and bacterial abundances were measured following Pfeilmeier et al. [79]. Briefly, plants were removed from clay, roots cut off and the whole phyllosphere part transferred into cold pre-weighed 2-ml tubes containing a sterile metal bead (5 mm diameter) and 200 $\mu$l 100 mM sodium phosphate buffer pH 7. Plant fresh weight was recorded, and plants were homogenized by shaking in the TissueLyser II (Qiagen) for 45 s at 25 Hz. After addition of 600 $\mu$l of 100 mM phosphate buffer pH 7 to homogenized plants, tubes were vortexed, and a ten-fold dilution series was prepared. To assess total pathogen colonization, 4 $\mu$l of the dilution series was spotted onto selective R2A agar containing 50 $\mu$g/ml rifampicin. Only one of the commensal strains (*Aeromicrobium* Leaf289) was resistant to rifampicin, though with delayed growth compared to the pathogen, making an unbiased assessment of the pathogen CFUs at 2 days post plating possible. To assess commensal colonization, 50 $\mu$l of the 10$^{-3}$ and 10$^{-4}$ dilutions were spread onto non-selective R2A agar supplemented with 0.5% methanol. Plates were incubated at room temperature until CFUs could be counted (2 to 7 days). The CFUs of the Mini5SynCom strains were counted separately based on colony morphology, the pathogen CFUs were not counted on these plates. Strains that were not detected were recorded as 0.09 at lowest dilution counted, to distinguish the numerical entry from detected ones and to rather overestimate colonization, especially for Pst. When two strains could not be distinguished, 20 colonies were randomly picked and identified by either restreaking on selective R2A agar with 0.5% MeOH (antibiotics added at the following concentrations in $\mu$g/ml: kanamycin (50), tetracycline (10), ampicillin (100), colistin (10)) or minimal medium agar containing methanol as sole carbon source[53], or by DNA fingerprinting using the Enterobacterial Repetitive Intergenic Consensus (ERIC) sequences protocol[82]. For ERIC-PCR, 25 $\mu$l reactions were set up containing 12.5 $\mu$l DreamTaq Green PCR Master Mix (Thermoscience, Cat.no. K1081), 6.5 $\mu$l distilled water, 2.5 $\mu$l of each 10 $\mu$M primer (ERIC1R (3′-CACTT AGGGGTCCTCGAATGTA-5′) and ERIC2 primer (5′-AAGTAAGTG ACTGGGGTGAGCG-3′)[82]), and 1 $\mu$l heat-lysed bacterial cells. PCR were performed in a thermocycler (Biometra, T1 thermocycler) with an initial denaturation (95 °C, 5 min), followed by 35 cycles of denaturation (95 °C, 30 s), annealing (50 °C, 1 min) and extension (65 °C, 2 min) with a single final extension (65 °C, 8 min). PCR products were separated on a 1.5% (w/v) agarose gel (Euroclone, Cat.no. EMR010500) supplemented with Gelred (Biotium, Cat.no. 41003) for 2 h at 80 V and patterns compared to known strains. Some commensal strains could still not be distinguished and were thus entered into the datafile as "ambiguous". The CFUs of those undistinguishable groups of commensals were taken into consideration in the calculation of the abundance of the Mini5SynComs strains. Total commensal colonization was calculated as the sum of the colonization of individual strains in their respective Mini5SynCom, the pathogen CFUs were excluded from that sum. Final measurements of bacterial colonization were in CFU per gram of plant fresh weight.

## Synthetic community assembly and pilot experiment

To test whether random communities of five strains have variation in pathogen colonization, random communities were designed by picking five strains at random without replacement using a webtool randomizer (www.randomizer.org) in a pilot experiment. As a strain pool, the SynCom-137[79] was chosen, which includes one representative strain per ASV of the *At*-LSPHERE collection[53]. Each of the 17 random communities was inoculated onto plants of four microboxes (*n* = 16), as described above, to control for box-to-box variation. For the pilot experiment, controls included plants in eight microboxes that were mock-inoculated with buffer instead of inoculation with synthetic communities. At infection timepoint, four of these eight microboxes were mock-infected with 10 mM $MgCl_2$ (axenic non-infected plants), while four microboxes were sprayed with the pathogen (axenic infected plants).

## SynCom size and strain pool size and composition for Mini5SynCom screen

The Mini5SynCom screen consisted in a plant protection assay with axenic *A. thaliana* Col-0[55] plants inoculated with randomly assembled synthetic communities (SynComs) of *At*-LSPHERE bacterial strains[53], and infected with a low dose of Pst, as previously established[20,22,45] (Supplementary Fig. 1a, b). We aimed to have each strain used in the screen present in about 20 communities on average to gain enough power in statistical analyses. For space and time reasons, we were constrained to a maximum of 136 microboxes dedicated to the screen of the Mini5SynComs (also see section "Synthetic community assembly and controls of the Mini5SynCom screen"), with each microbox dedicated to one random community. Based on the pilot experiment, we decided for a SynCom size of five strains. We calculated the expected prevalence of a strain across all screened communities as

$$E(X) = \frac{k}{n} \times N \qquad (1)$$

with k being the size of the Mini5SynCom (k = 5), *n* the size of the pool of strains to choose from, and *N* the number of screened Mini5SynComs (N = 136). On the left hand of the multiplication sign, the equation corresponds to a simplification of the number of possible communities that share a specific strain $\binom{n-1}{k-1}$ divided by the total number of possible five-strain communities drawn from a n-strain pool $\binom{n}{k}$. We calculated that the optimal size of the pool was 35 strains or lower (Fig. 2a). Strains to be included in this 35 strains pool (SynCom-35) were selected from the SynCom-137[79], which includes one representative strain per ASV of the *At*-LSPHERE collection[53]. First, two Mini5SynComs were chosen from the pilot experiment based on pathogen luminescence and strain diversity, to control the reproducibility of each experimental round. Mix6 was chosen for its consistently high pathogen luminescence (renamed SynCom-High), and Mix12 for its consistently low pathogen luminescence (renamed SynCom-Low) (Supplementary Fig. 2). Then, to bring the strain pool size from 10 strains to 35, additional strains were selected to reflect phylogenetic diversity of the *At*-LSPHERE[53], to include diverse levels of individual protection[45] and to generate overlap of strains used in prior studies (Fig. 2b; Supplementary Data 2; Supplementary Fig. 1c)[55,75,76].

## Synthetic community assembly and controls of the Mini5SynCom screen

Random communities of the random Mini5SynCom screen experiments were designed using a webtool randomizer (www.randomizer.org), which randomly picked five strains per SynCom without replacement from the SynCom-35 strain pool (see section "SynCom size and strain pool size for Mini5SynCom screen"). The Mini5SynCom screen was partitioned into two experimental rounds (Experiment 1 and 2), consisting of 80 microboxes each, which is the maximum of

microboxes that fit into one plant growth chamber (CU-41L4, Percival). For each experiment of the Mini5SynCom screen, following controls were included: one box inoculated with the SynCom-Low and another with the SynCom-High (two SynComs characterized by low and high pathogen colonization in the pilot experiment), one inoculated with a community comprising all 35 strains of SynCom-35 and nine microboxes that were mock-inoculated with buffer, two of which were mock-infected with buffer instead of pathogen at the infection timepoint (axenic non-infected plants). This resulted in a total of 136 microboxes to screen random Mini5SynComs. For the independent test set (experiment 3), the same number of boxes were used as for the screen (68 random Mini5SynComs, three control communities, two axenic non-infected and nine axenic infected controls). Plant images at 14 dpi of the Mini5SynCom screen are available (see "Data availability").

## Validation experiment of machine learning results

We empirically tested the ability of strains identified in machine learning to reduce pathogen colonization, *i.e. Pseudomonas* Leaf15, *Rhizobium* Leaf68 and *Acidovorax* Leaf76, and additionally *Rhizobium* Leaf371 (Fig. 6). The experiment was split onto two experimental rounds (Replicate 1 and 2), in total each strain was inoculated to 24 plants (six microboxes, four plants each) and later infected with Pst (see "Plant infection" above). To test for potential synergic effects, each binary combination of these strains and all four strains together were inoculated onto 24 plants equally partitioned between the two experimental rounds. To demonstrate that the identified strains provided better protection than other strains, same treatment scheme was applied to four randomly selected strains as additional controls (*Sphingomonas* Leaf21, *Dyadobacter* Leaf189, *Rhodococcus* Leaf225, *Frigoribacterium* Leaf254). We compared the pathogen colonization among those treatments, 40 axenic plants infected with Pst, and 24 plants inoculated with SynCom-35 (also equally split between two experimental rounds). Each experimental round of the validation experiment included following control conditions: three boxes for each control community (SynCom-Low, SynCom-High and SynCom-35 (see previous section)), 3 axenic non-infected boxes, 5 axenic infected boxes. Plant images at 14 dpi are available (see "Data availability").

## Validation experiment of synergic effect of strains

After further exploration of data (Fig. 8a–c), we raised the hypothesis that two strains, *Rhizobium* Leaf371 and *Arthrobacter* Leaf337, could have a synergic effect on pathogen reduction. To empirically test this hypothesis, we inoculated each of these strains and their combination on a total of 24 plants split onto two experimental rounds (Round 1 and 2). In addition, we included three Mini5SynComs containing the two strains of interest together. One of these Mini5SynComs was taken from the original screen (Mix6) and two validation mixes (new Mini5SynComs) were assembled with three random non-PR strains (ValMix1 and 2). In addition to the two strains of interest, Mix6 was composed of *Sphingomonas* Leaf32, *Brevundimonas* Leaf168 and *Curtobacterium* Leaf183, ValMix1 of *Methylobacterium* Leaf122, *Rathayibacter* Leaf299 and *Methylobacterium* Leaf466, and ValMix2 of *Exiguobacterium* Leaf187, *Microbacterium* Leaf203, and *Aeromicrobium* Leaf289. Each experimental round of the validation experiment included following control conditions: three boxes for each control community (SynCom-Low, SynCom-High and SynCom-35 (see previous section)), three axenic non-infected boxes, five axenic infected boxes. Plant images at 14 dpi are available (see "Data availability").

## Data analysis

**Data transformation and exclusion.** Data were analyzed with R 4.0.5[83]. All colonization (CFU) and luminescence measurements were $log_{10}$-transformed prior to analysis. Where mentioned in the text, pathogen colonization was normalized within experimental round by subtracting the median of axenic infected control samples. Any data

point not included in the closed interval [Q1 − 1.5 IQR, Q3 + 1.5 IQR] was flagged as an outlier, with IQR being the interquartile range. We excluded plants which showed fungal contamination, and among outliers the samples that were disturbed during experimental procedure (i.e., the microbox fell down).

**Pilot Experiment.** Luminescence measurements were used as a proxy for pathogen colonization at 3, 6 and 12 dpi as described previously[45], which was enabled by using a *luxCDABE*-tagged derivative of the pathogen[80]. Briefly, microboxes were placed with open lid into the IVIS Spectrum Imaging System (Xenogen) and luminescence was acquired for 30 s with a 490-510 nm emission filter. In the Living Image Software v.4.2., circular regions of interest (ROIs) were set around each plant, adjusting to bigger plants if necessary, and exporting the total photon flux per ROI. Prior to data analysis, the total flux [p/s] measurements were $\log_{10}$-transformed. Differences in luminescence among random communities and among boxes inoculated with the same community were detected using Welch's ANOVAs (Source Data, Supplementary Data 1, Supplementary Fig. 2, 3). To test whether pathogen luminescence can be detected for the different treatments (*i.e.*, luminescence higher than background luminescence of axenic non-infected plants), we performed one-sided Welch's t-tests. Based on the results from the pilot experiment (see Results), we finally used CFU counting technics to estimate pathogen colonization for the Mini5SynCom screen and validation experiments.

**Calculating phylogenetic diversity in Mini5SynCom screen.** To calculate the phylogenetic diversity of Mini5SynComs in the screen, we constructed a phylogenetic tree using the draft-genomes of the *At*-LSPHERE strains[45,53] and the de novo workflow of GTDB-Tk[84] v2.3.0. The procedure consists in an alignment of the *At*-LSPHERE draft genomes with the GTDB-Tk reference genomes, and a maximum-likelihood tree inference conducted with FastTree[85] with WAG + GAMMA models. Because the position of the root in the bacterial tree remains uncertain, we used metrics which can be calculated from unrooted phylogenetic trees. For the presence/absence data, we used Faith's phylogenetic index[86], and to account for the abundance of strains in communities, we calculated the weighted mean pairwise distances (mpd) using the package Picante[86] v1.8.2.

**Correlation of pathogen colonization with commensal colonization, community evenness or phylogenetic diversity in Mini5SynCom screen.** We modelized the colonization of Pst (dependent variable) in experiments 1 and 2 with three sets of generalized mixed models (Source Data). The fixed effects were commensal colonization, Mini5SynCom evenness (Pielou's index) or phylogenetic diversity (see above), respectively. Each group of models included a full version with random intercepts and/or slopes for experiments and microboxes, all nested random structures, and a model with no random effects. When both present, the microbox effect was nested in the experiment effect. Furthermore, microbox grouping was completely confounded with the composition of Mini5SynComs. For each set, we calculated the Akaike information criterion (AIC) with the base R function "AIC" to select the best models (delta AIC < 4) and present the results based on these. We excluded from these analyses all controls and problematic outliers (see above) to only include Mini5SynComs. Because the calculation of the evenness required the abundance of each member of communities, we also excluded Mini5SynComs with ambiguous values, and conducted a sensitivity test, which consisted of excluding all communities with some strains being below level of detection. At the end, 134 and 122 Mini5SynComs were included for the regression of Pst abundance with the abundance and evenness of communities, respectively. Because we did not include any sample with no commensal in analyses, we centered the commensal colonization before analyses to interpret the intercept as the expected Pst colonization at mid-levels of commensal colonization. For the phylogenetic diversity, we conducted analyses with the Faith's diversity index calculated from inoculum compositions to account for the initial diversity inoculated to the plant. We reproduced this analysis with Mini5SynComs having all five strains unambiguously detected at the end of the experiment. To analyse the link between the realized diversity and Pst colonization, we calculated the weighted mpd of communities with all Mini5SynComs without ambiguous abundances at the end of the experiment. As for the evenness, we also conducted a sensitivity test which consisted of excluding all communities with strains being below level of detection. All models were fitted with the restricted maximum likelihood method with the function lmer when random effects were present (package lme4 v1.1.28[87]) or the function gls for models without random effects (package nlme v3.1.157). We examined the response plots, residual distributions, and residual plots for no deficiency patterns in the fit of models (Supplementary Fig. 4).

**Machine learning training of algorithms.** We trained random forests (RF) and elastic-net regularized generalized linear models (GLMNet) to predict pathogen outcomes based on the 35 strains (*i.e.*, features) making up the Mini5SynComs (Supplementary Fig. 5). The 136 randomly assembled Mini5SynComs screened in experiment 1 and 2 of the screen served as training data (the controls were not used) (Source Data). For classifications, two classes were made−"protected" and "non-protected"−based on the global minima present in the bimodal distributions of the pathogen colonization of experiments 1 and 2 (Fig. 6a, b) suggesting the presence of two groups with low and high pathogen colonization, respectively (Supplementary Data 5). To estimate if these local minima were not the result of stochastic events during population sampling, the datasets of the two experiments were bootstrapped 1000 times and measured the relative frequency of detection of local minima in the same regions of the density curves (Supplementary Data 4). After this control, we classified samples into a "protected" class (pathogen colonization lower than the minima, positive class) and a "non-protected" class (pathogen colonization equal or higher than the minima, negative class) for each distribution. For regression analyses, we predicted the pathogen outcomes based on the normalized pathogen reduction of a Mini5SynCom by subtracting the mean pathogen colonization of the axenic control in each experiment. We trained the models on individual plant measurements (544 plants) as well as on aggregated pathogen abundance data (median pathogen CFU/g for one Mini5SynCom; 136 microboxes). Predictors were either presence/absence of Mini5SynCom members or the absolute abundance of these determined by CFU enumeration at the end of the experiment ("colonization") (Source Data). When strains could not be unambiguously assigned, the recovered CFU were partitioned equally between the non-distinguishable strains (16% of samples (88 plants), or 6.7% of inoculated strains). Models were fitted using the R pacakge caret[88] with the embedded models randomForest[89] in conjunction with e1071[90] and GLMnet[91], for RF and GLMnet, respectively. All model / algorithm combinations (12 in total; Supplementary Fig. 5) were tuned using repeated k-fold cross-validation using 10 rounds of 5-fold cross-validation. When individual plant measurements were used instead of aggregated data, measurements were stratified to ensure that all samples of a Mini5SynCom were either in the training or validation set to avoid overfitting. The metrics used to select the best tuning parameters were kappa and root-mean-squared errors for classification and regression, respectively. Final models were then fitted to all training data using the tuned parameters. Because pseudo-random processes were involved in those procedures, we integrated all results from the repetition of all analyses with eight different starting seeds.

**Performances of trained algorithms.** To minimize data leakage leading to overestimation of model performances, we used an independent test set to assess the quality of our trained classifiers and regression models. To do so, we conducted a separate experiment (experiment 3) following the same procedures as for experiments 1 and 2, with 68 new Mini5SynComs randomly generated from the same pool of strains and including the same controls (Source Data). Because the two control communities SynCom-High and SynCom-Low had not been used in the training of the different models, we included these controls in the test set, increasing the number of Mini5SynComs in the test set to 70. The distribution of pathogen colonization was comparable to the training dataset, and its bimodal shape was not sensitive to perturbation across 1,000 bootstrap replicates (Supplementary Fig. 6d, Supplementary Data 4), so the same classification approach was taken. For classification, the performance metrics of trained models were compared with a random classification of the test samples (Monte Carlo simulation) (Supplementary Data 6). For regression analyses, we compared the performances of the trained models with performances obtained from constant prediction of the mean pathogen colonization of the entire dataset (No model) (Supplementary Data 8). Because we were interested in finding the strains being most important to predict plant protection, we extracted the relative importance of strains from all ML analyses using the varImp function implemented in the R library caret (Supplementary Data 8). We verified the sensitivity of this relative importance to the uncertainty on the position of minima used in the training set. We re-fitted the RF classifier with presence / absence of strains and individual plant measurements after reclassification of training samples using ten different values of minima sampled randomly between the 5th and 95th percentiles of the bootstrapped distributions of local minima of experiments 1 and 2 (see above) (Supplementary Data 9).

**Validation experiments.** To test for significant differences in pathogen colonization in the validation experiment of the strains predicted to be most important for machine learning and the strain combination, we used generalized mixed models after averaging the pathogen colonization per microboxes to avoid grouping structure in the data (Source Data). We followed the same procedures as described in section "Correlation of pathogen colonization with commensal colonization, or community evenness or phylogenetic diversity in Mini5SynCom screen" for the fitting of models, model selections, and diagnostics of fit. All models included the inoculation treatment as a fixed effect with or without random effects. The most complete random structure included random intercepts and slopes for experimental runs. The significance level of contrasts was 0.05 after Bonferroni correction. The families of hypotheses used for those corrections were all binary-combinations of contrasts included in Figs. 7a, b, and 8d to keep the probability of one or more false positive inferior to 0.05 in each corresponding sections of the manuscript.

### Reporting summary
Further information on research design is available in the Nature Portfolio Reporting Summary linked to this article.

## Data availability
The original images of plants at harvest timepoint (14 dpi) have been deposited on Zenodo and are available under https://doi.org/10.5281/zenodo.8399345. Source data generated in this study are provided with this paper and its supplementary information files and data.

## Code availability
The source code generated in this study has been deposited on Zenodo and is available under https://doi.org/10.5281/zenodo.10118600.

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

## Acknowledgements

We thank Christopher M. Field for the provision of the phylogenetic tree used to calculate the phylogenetic diversity in this study. We thank Pascal Kirner, Martin Schäfer, Lucas Hemmerle, Sebastian Pfeilmeier, Trisha Stewart and Donika Demaj for their support during experimental procedures. We thank Mikolaj Rybinski for helpful discussions and support during the screen parameter design. This study was funded through a European Research Council Advanced Grant (PhyMo, grant number 668991) and the NCCR Microbiomes funded by the Swiss National Science Foundation (grant number 51NF40_180575) to J.A.V.

## Author contributions

B.E., J.M, C.M.P and J.A.V. designed the study, B.E., C.M.P., B.A.M. and M.B.M. performed the lab work, B.E., J.M. and C.M.P. analyzed the data and conducted statistical analyses. B.E., J.M, C.M.P and J.A.V. wrote the manuscript.

## Competing interests

The authors declare no competing interests.
