## [Peer Review File · Nature Communications]

REVIEWER COMMENTS

Reviewer #1 (Remarks to the Author):

Remarks to the Author:

The manuscript by Emmenegger et al. used machine learning and validation tests to explore properties relevant to microbiota-conferred host protection in a synthetic community context. By screening 136 randomly assembled synthetic communities (SynComs) of five strains, the authors identified strain identity as the most important predictor of pathogen reduction. Validation experiments confirmed three strains as the main drivers of pathogen reduction, and two additional strains conferred protection in combination. This study provided a framework that can be adapted to identifying features relevant to microbiota function in other biological systems. Although the concept of applying SynComs to study the host phenotypes is not novel, this interesting manuscript provides a valuable approach by combining machine learning and validation test to obtain the optimal SynComs for studying the interaction between phyllosphere microbes-plant phenotypes. I agree with the contribution of this manuscript to the field, but some methods and conclusions that need to be clarified by the authors.

Major concerns:

My main concern about this manuscript is related to the method, the Results section contained many method contents, and the Method section was undetailed. For instance, how was the At-LSPHERE obtained? How many plant samples and what was the source microbes (air or soil); whether the 136 random synthetic communities were derived from 136 plant individuals or boxes and how Mini5SynComs was assembled. We suggested the authors add a Method sub-section on experimental design (including experiments 1, 2 and 3) to elaborate on the strain pools, SynComs, and Mini5SynComs mentioned in the text.

I also have somewhat major vague in the constructions of the regression and classification models, the authors should provide the constructed data matrix, including the output and input features. This will help readers to have a more intuitive understanding of the methods used by screening SynComs. Moreover, did the authors consider directly using the presence and absence of all isolated strains from *A. thaliana* Col-0 and the value of pathogen colonization to construct a machine learning model? and what is the difference compared with the method in this study?

Specific comments:

1. P1-Line 16: What is this potential? Functions?
2. P1-Line 21: What is the strain identity? taxonomy? please clarify.

3. Results: There are too many descriptions of the methods in the Results section.
4. Results: P values are in italics.
5. P3-Lines 20-24: Why the authors selected the five strains to construct SynCon? Did the authors conduct other experiments? If so, please provide data analysis or related references.
6. P3-Lines 26-27: How to select the strain pools? Were the 17 Mini5SynComs randomly assembled by using statistical screening? Please clarify.
7. P3-Line 35: Please use uniformly adjusted P values throughout the manuscript.
8. P4-Lines 11-12: As known, the phyllosphere microbiomes are mainly derived from soil dust, seed germination or atmospheric microbe. So, in this study, what is the source of the strain pools? Isolated from each plant? What determines the size of the strain pools?
9. P5-Line 23: The correlation relationship is not uniform due to the difference in the order of magnitude, does it mean there is no relationship between them? because the difference in the order of magnitude does not change the trend of correlation, positive or negative.
10. P5-Line 23: Did the 70 random Mini5SynComs come from the 136 strain pools?
11. P6-Line 38: Are the RMSE values greater than 1? Please clarify.
12. P7: Did the authors consider which other two strains had better pathogen reduction when combined with Acidovorax Leaf76, Rhizobium Leaf68, and Pseudomonas Leaf15? Furthermore, by screening different SynComs, the authors finally obtained the three biomarker beneficial bacteria. How is this different from screening beneficial bacteria directly based on plant phenotype using machine learning? In other words, what are the advantages in the method of this study?
13. P10-Lines 7-8: Is the interaction of the three strains in the community synergistic growth?
14. P10-Line 34: Are the growth chambers strictly sterile?
15. P11-Line 1: How many boxes were used in this experiment?
16. P11-Line 6: Please provide the full name of "At-LSPHERE".
17. P12-Line 21: How to screen the Mini5SynComs? Please added references or related details.
18. P14-Line 1: Did perform normal distribution and homogeneity analysis of all data before using ANOVA and t-test analyses?
19. P16-Line 6: Please remove the misadded brackets.
20. Figures: The first letter of the Y-axis and X-axis label is capitalized. Please revised throughout this manuscript.

Reviewer #2 (Remarks to the Author):

The authors demonstrate an approach to identify individual bacteria that robustly affect plant health in a community context. Specifically, they used random synthetic communities of bacteria to evaluate protection against a bacterial leaf pathogen, then used machine learning to identify strains that were strongly associated with protection against the pathogen. Most impressive was their very thorough validation, including 68 independent syncoms to validate the machine learning, thorough experiments to validate the protective strains, as well as experiments identifying other protective strains, which further showed that even slightly less “predictive” strains identified with machine learning were protective in combination. I think this is a novel approach until now, I really enjoyed the study and think that it could inspire a lot of interesting work in the future.

A couple of comments are minor but are for me most important:

Based on Figure 8a. I could be wrong, but it seems like if one just had identified the clear bimodal distribution, then manually looked at what strains were most frequent in the “protective” syncoms that were not in the non-protective ones, one would likely find the same strains without complicated machine learning approaches that require special knowledge and extensive validation. Despite that, I think this is an important demonstration of an approach that can bring clear advantages. The authors could more clearly discuss in what sorts of screens this approach would really outperform manual analysis, i.e., when would manual analysis become unrealistic – with large pools, more complex communities, or screening for combinations of taxa or more complex traits, maybe even multiple traits perhaps?

Another point that I think might be important is that learning models are very common for predictions in complex systems and this is an adaptation of these approaches. I think some literature and discussion of this is missing. This includes fairly common use of for example random forest models in plant microbiome studies, mostly to identify taxa linked to specific plant organs, sampling sites, etc. The work here is clearly a distinct application from that, but still such studies could be cited in the discussion of what makes this work unique.

Further minor comments:

Intro

L9: “..., and ... - or...”

Results

First section:

I think this section is a very nice overview of the considerations that go into the study design, which is very nice. However, in my opinion it distracts from the rest of the story, as other syncom sizes, pool sizes, etc. are ultimately not used. Thus in my opinion it could be supplemental information.

P3, L12: It struck me that the infection was allowed to go so long, which is unusual. I see in the methods that this was a very low inoculation dose, so that makes sense. Maybe it could be mentioned here that this was a low-dose infection to avoid confusion.

In figure 2 and P3, L13-15, SynCom size is clearly defined as number of strains but Strain pool size is not really clearly defined. I thought first total number of cells based on Fig 2a, but this would not affect strain frequency, so it must be something else. Note: after reading on I realize the pool is referring to the pool of strains drawn from (Fig 2b) and the frequency is how often the strains show up in communities. But this was not clear to me until later.

P3, L35: I don't really have a problem with using CFUs (see also comments below), but I have trouble following the reasoning here. Are you essentially saying that luminescence is too insensitive? That is important, so perhaps that could be more explicit.

P4, L20: Is there a reason that the strain frequency plot in fig 2a has no axis scales?

I think it would be nice for readers throughout the manuscript if Fig2b already had some more detailed taxonomy, like genus names.

Second section:

Methods related to pathogen CFU counting and Fig 3a: In our experience, on R2A many bacteria are look very similar and Rif-resistance occurs among leaf-colonizing bacteria. Do the authors have any assurance that DC3000 was not at least occasionally over-counted because of the other strains in the mix?

Methods related to commensal CFU counting and Fig3b: It is not completely clear to me - the "commensal" counts were done on non-selective media, so do they include DC3000, or was it always

excluded or its counts on selective media subtracted from totals? Perhaps that could be mentioned more clearly in the results/methods.

Fig 3: The figure header says there is a scatterplot overlaid, but I cannot see it.

Third section:

P5, L19-20: If the pathogen is included in the commensal levels (see previous comment) would that explain this positive correlation?

Fig 4: In the header, “Fixed effect in the model” – maybe referring to the details in the methods section would help the figure/heading stand alone better.

Fourth section:

I realize the methods of this part are very important for the results, but it should be noted that this section reads a lot like methods. I’m not really sure how much this can be improved upon, but I think some details could be relegated to the methods section.

The verification of the machine learning models is really nicely done with an independent data set. However, it was not clear to me if the authors checked how many of the 68 new Mini5Syncoms were already represented in the original set – I would think that it would be important for them to be new combinations to really check the predictions.

Fig 5: The meaning of “presence/absence” and “colonization” groups are not clear from the figure heading. RF and GLMNet are not in the legend either.

P6, L17: Are abundances referring to counts at the end of the experiment, I guess?

P6, L29-32: I am not an expert on this, but I think precision is true positives / (true positives + false positives) and recall is true positives / (true positives + false negatives). I think the way the definitions are written here would be identical or the difference hasn’t been made clear.

Fifth section:

The use of random strains and combinations is a really nice control for this experiment, well done.

P7, L33: This is true, but also the other pairs seem to be comparable.

Sixth section:

Fig8: ValMix and Mix are not clearly explained in the figure heading

Discussion:

The authors do a very nice job of discussing the results. Throughout, I was wondering if machine learning could be implemented to identify combinations, besides the manual identification used here. The authors do touch on this in L28-31. Maybe they could go a bit deeper, for example I note that in the PR Strains mixes (Fig 8a), there is a very large variation over 4 orders of magnitude – can this be used further to identify “helper” strains or others that negatively affect the effectiveness of a useful strain, besides just combinations of the most predictive ones?

Looking at Supp Fig2c, most results agreed with Vogel et al., but some not (e.g., Acidovorax 76 was not protective there and Strain 21 was not very predictive of protection here). Maybe the authors could briefly discuss what might cause the differences.

Reviewer #3 (Remarks to the Author):

The authors aimed to identify the beneficial microbiome that can protect plant health. They used experimental and machine learning methods to randomize and simplify the synthetic communities (SynComs), and then examined the disease resistance of these SynComs. The study provides strong evidence for the ability of SynComs to enhance host adaptation. Additionally, the framework proposed in this paper is important for efficiently constructing synthetic communities in future studies. However, there are a few issues that should be addressed. Notably, the absence of high-throughput sequencing data, plant resistance gene response data, and quantitative data from pathogen qPCR analysis in this paper is significant. Including these components is crucial for reaching a more reliable conclusion. Due to these limitations, I have significant reservations about the findings of this paper. Please refer to my detailed review comments below.

1. The introduction lacks coherence and fails to provide a comprehensive overview of the development of the synthetic microbiome and its current challenges. Furthermore, it does not clearly state a well-defined scientific question.

2. The manuscript would benefit from including amplicon data to track the dynamics of microbial communities. It is also important to include negative controls and conduct amplicon sequencing to determine if the SynComs were affected by contamination from environmental bacteria and fungal spores.
3. The author should provide detailed information regarding the machine learning analysis, including the missing error rates and Tenfold cross-validation curves.
4. Extracting a cohesive conclusion from this manuscript, which incorporates all the presented data, is challenging. This difficulty primarily arises from shortcomings in the experimental design, such as the absence of microbial amplicon data, the lack of a negative control in unplanted soil samples, and the omission of plant-resistant gene/pathway-related data.
5. The evidence or data supporting the selection of the five bacteria for the SynCom is not sufficient, although it is mentioned that they were obtained from a previous pool of 137 strains.
6. The excessive use of the active voice has negatively impacted the readability of this article.
7. Relying solely on the dilution smear plate method for quantitative analysis of synthetic communities and pathogenic bacteria poses uncertainties, as this method is relatively unreliable and subjective. I strongly recommend incorporating second or third-generation high-throughput sequencing for quantitative analysis and dynamic tracking, which will provide more reliable verification of abundance changes in the SynComs and enable confirmation of whether the selected three bacteria are indeed the core bacteria.
8. It is necessary for the author to include qPCR and RNA-seq data from plants to substantiate the disease-resistant ability of the synthetic communities (SynComs). It is also crucial to perform absolute quantification of pathogenic bacteria using qPCR.
9. There is a lack of useful information in the major Figures, such as Figure 1, Figure 2, Figure 3, and Figure 4.
10. Furthermore, there is no explanation of the pathogenicity index or provision of images related to diseased plants.
11. Furthermore, exploring genes involved in plant-phyto microbiome signaling pathways, such as those responsible for LCOs and bacteriocins, and investigating their response to pathogens would be valuable.
12. In the "Strain pool size for community screen" section of Methods, why should each strain appear in 20 communities and not more? In addition, $E(X) = [k!(n-1)! / n!(k-1)!] * N$, isn't that equal to $(k/n)*N$?
13. The paragraph 3 of introduction, which introduces the *Arabidopsis thaliana* phyllosphere is good material for exploring these questions. But it is not coherent to move directly to machine learning. The background on machine learning for mining beneficial microorganisms is not well explained. The key questions addressed by the study are not well expressed.
14. Whether the synthetic community developed in this study to identify beneficial microorganisms is applicable to other plants? This could be discussed.

15. The significance test symbol (abcd) in Figure 8d may be incorrect? and the "Bonf. corrected p-value" in Table S11, S12 and S15 should not be greater than 1.

16. Reference 13 not following journal style?

REVIEWER COMMENTS

Reviewer #1 (Remarks to the Author):

Remarks to the Author:

The manuscript by Emmenegger et al. used machine learning and validation tests to explore properties relevant to microbiota-conferred host protection in a synthetic community context. By screening 136 randomly assembled synthetic communities (SynComs) of five strains, the authors identified strain identity as the most important predictor of pathogen reduction. Validation experiments confirmed three strains as the main drivers of pathogen reduction, and two additional strains conferred protection in combination. This study provided a framework that can be adapted to identifying features relevant to microbiota function in other biological systems. Although the concept of applying SynComs to study the host phenotypes is not novel, this interesting manuscript provides a valuable approach by combining machine learning and validation test to obtain the optimal SynComs for studying the interaction between phyllosphere microbes-plant phenotypes. I agree with the contribution of this manuscript to the field, but some methods and conclusions that need to be clarified by the authors.

We thank the reviewer for the summary and the positive assessment.

Major concerns:

My main concern about this manuscript is related to the method, the Results section contained many method contents, and the Method section was undetailed. For instance, how was the At-LSPHERE obtained? How many plant samples and what was the source microbes (air or soil); whether the 136 random synthetic communities were derived from 136 plant individuals or boxes and how Mini5SynComs was assembled. We suggested the authors add a Method sub-section on experimental design (including experiments 1, 2 and 3) to elaborate on the strain pools, SynComs, and Mini5SynComs mentioned in the text.

The At-LSPHERE strains were previously isolated from healthy-looking environmental Arabidopsis plants (ref 53). We have streamlined the results section and now provide more details on the experimental design, strain pool, and Mini5SynComs in the Materials and Methods section to improve clarity.

I also have somewhat major vague in the constructions of the regression and classification models, the authors should provide the constructed data matrix, including the output and input features. This will help readers to have a more intuitive understanding of the methods used by screening SynComs. Moreover, did the authors consider directly using the presence and absence of all isolated strains from *A. thaliana* Col-0 and the value of pathogen colonization to construct a machine learning model? and what is the difference compared with the method in this study?

We now documented all input and output matrices and features in the Supplementary Material (Supplementary Tables 4-9 and Supplementary Data 1 of the revised version). We refer to these sources on pages 17-19.

Presence and absence of strains were used as explanatory variables in our machine learning models (p. 6, l. 14-15): “As predictors (features), we used presence/absence or absolute abundances of Mini5SynCom members (“colonization”).” Because we needed to ensure a minimal prevalence of the strains in the randomly assembled Mini5SynComs, we did not use the entire At-LSPHERE collection in machine learning analyses. However, we agree that screening could be adapted in the future for a broader collection of strains (see Figure 2a for a projection of the number required).

Specific comments:

1. P1-Line 16: What is this potential? Functions?

By “potential”, we mean “contribute to host phenotypes”. We now rephrased the abstract to clarify the perspective of our study and avoid this ambiguity. It now reads: “Plant-associated microbiomes contribute to important ecosystem functions such as host resistance to biotic and abiotic stresses.”

2. P1-Line 21: What is the strain identity? taxonomy? please clarify.

“Strain identity” refers to a genetic variant in bacterial species, and as such is a subspecies taxonomical group. The taxonomy of the strains used in this study is referred to on p. 4 l. 14, visually represented in Figure 2b, and documented in Supplementary Table 2.

3. Results: There are too many descriptions of the methods in the Results section.

As suggested by the reviewer, we transferred some information on the methods from the results to the method section.

4. Results: P values are in italics.

Corrected.

5. P3-Lines 20-24: Why the authors selected the five strains to construct SynCon? Did the authors conduct other experiments? If so, please provide data analysis or related references.

The choice was based on a previous study showing that communities between 3 and 10 strains can generate variation in pathogen colonization (ref 45). It was also based on our expectation that we would be able to distinguish all strains on plates (p. 3 l. 26.). We had tested SynComs of 5 in a pilot experiment in the present study (p. 3 l. 28-32, Supplemental Figure 2). We modified the corresponding paragraph to improve clarity and the basis of our reasoning (p. 3 l. 21 - p. 4 l. 2).

6. P3-Lines 26-27: How to select the strain pools? Were the 17 Mini5SynComs randomly assembled by using statistical screening? Please clarify.

For the pilot experiment, we used a randomizer tool to randomly pick and assemble the 17 Mini5SynComs from 137 strains of the At-LSPHERE that covers the entire diversity of the strain collection based on ASV assessment (ref 79). We rephrased the corresponding section to improve clarity and provide more details in the Methods section (p. 13 l. 23-25).

7. P3-Line 35: Please use uniformly adjusted P values throughout the manuscript.

While we had adjusted p-values using Bonferroni method throughout the manuscript, we had deliberately made an exception for the pilot experiment and chose to be conservative and not apply a p-value correction. Indeed, for this specific analysis, p-value correction would have further supported our final decision of using CFU counting rather than luminescence, as more communities will have luminescence that is not significantly different from background.

8. P4-Lines 11-12: As known, the phyllosphere microbiomes are mainly derived from soil dust, seed germination or atmospheric microbe. So, in this study, what is the source of the strain pools? Isolated from each plant? What determines the size of the strain pools?

All strains were isolated from leaves of environmentally grown *Arabidopsis* plants, as described in reference 53. We rephrased the introduction of the At-LSPHERE strain collection “environmentally representative collection” (p. 2 l. 31-34). We agree these bacteria were colonizing the plants from different

environmental sources before we had isolated them from leaves. The size of the strain pool was an important component because it conditions the average prevalence of strains across Mini5SynComs. We rewrote the first sentence of this paragraph to make this point clearer (p. 4 l. 10-12): “The size of the strain pool [...] that determined the prevalence of each strain across Mini5SynComs.”

9. P5-Line 23: The correlation relationship is not uniform due to the difference in the order of magnitude, does it mean there is no relationship between them? because the difference in the order of magnitude does not change the trend of correlation, positive or negative.

The reviewer is right, the strength of the relationship between commensal and pathogen colonization changes across communities. We took this into account by estimating the sampling variance of the estimate, and accounting for this variation in the test showing that this strength is significantly different from zero. For this reason, we can conclude that despite variation across communities, there is an overall positive relationship between commensal and pathogen colonization. We modified the text to clarify that despite variation in strength across communities, the presence of a positive relationship between commensal and pathogen colonization was significant (p. 5 l. 7-10): " For overall commensal colonization, the four best models ($\Delta AIC < 4$) supported a significant increase in commensal colonization of one order of magnitude with an equivalent increase in pathogen colonization considering all analyzed Mini5SynComs (Fig. 4a; Supplementary Table 3)".

10. P5-Line 23: Did the 70 random Mini5SynComs come from the 136 strain pools?

We think this comment refers to Page 6, Line 23. The 70 random Mini5SynComs are an independent and unique dataset, none of which have the same compositions as any Mini5SynCom used in the training dataset. They were randomly assembled from the same pool of 35 bacteria (SynCom-35) which was used in the training. We clarified the uniqueness of the test dataset (p. 6 l. 20-21): “Subsequently, we evaluated the performance [...] a new set of 70 random Mini5SynComs, none of which had a composition present in the training sets.”

11. P6-Line 38: Are the RMSE values greater than 1? Please clarify.

Indeed, the RMSE values can be above one as they are calculated from residuals. In our case, the RMSE of trained models ranged from 0.79 to 1.06 (p. 6 l. 35): “with a root-mean-squared error (RMSE) ranging from 0.79 to 1.06.” The RMSE of predictions using the global average of data, was 1.5 (p. 6 l. 36): “compared to an RMSE of 1.5 for the global average.”

12. P7: Did the authors consider which other two strains had better pathogen reduction when combined with *Acidovorax* Leaf76, *Rhizobium* Leaf68, and *Pseudomonas* Leaf15? Furthermore, by screening different SynComs, the authors finally obtained the three biomarker beneficial bacteria. How is this different from screening beneficial bacteria directly based on plant phenotype using machine learning? In other words, what are the advantages in the method of this study?

The combinations of the three strains, i.e. *Acidovorax* Leaf76, *Rhizobium* Leaf68, and *Pseudomonas* Leaf15, already reduced pathogen colonization to the level of the control SynCom-35 (i.e., the most protective communities in the present study). We therefore did not investigate further which potential other two strains would have led to better pathogen reduction when added to the three most protective strains. Also, this investigation would have meant to test another 595 combinations with several replicates each, which would have made the scale of the experiment unfeasible.

Regarding testing for plant phenotype, in principle, this could be used as the dependent variable in machine learning analyses. However, it presents some challenges which we think will complicate the

suggested procedure. In our experimental readouts, pathogen numbers provide a higher level of variation and unbiased way for quantification. Scoring plant phenotypes would require an advanced image segmentations technics (*e.g.*, deep learning approaches) to rapidly translate phenotypes, such as leaf damages, into unbiased and accurate metrics allowing detection of statistical associations. Manual examination and scoring could be conceivable, but present other type of potential difficulties such as the handling experimenter biases, and non-trivial standardization procedures. We define protection as colonization resistance in Page 3 line 3. To be transparent and share all images of plants, we now provide them on zenodo, see data availability, including link to <https://doi.org/10.5281/zenodo.8399345>.

Regarding the advantages of our method compared to alternative approaches, we now included a paragraph about this aspect in our discussion (p. 10 l. 1-25).

13. P10-Lines 7-8: Is the interaction of the three strains in the community synergistic growth?

Because we investigated infected plants only, and infection status can bias commensal growth, we prefer not to draw definite conclusions from the commensal data. However, we agree this is a valid point. We added a discussion on whether the PR strains can enhance the growth of each other (p. 10 l. 35).

14. P10-Line 34: Are the growth chambers strictly sterile?

The growth chambers are not sterile, but all plants were inside microboxes which ensure sterile and axenic conditions without limiting gas exchanges (ref 53, 55, p. 11 l. 23-32).

15. P11-Line 1: How many boxes were used in this experiment?

For the initial screen, we used 136 boxes total with additional boxes for control communities (p. 14 l. 3). Each random community was inoculated on four plants per box (see Material and Methods; p. 11 l. 30). The test data set was generated from 68 boxes plus controls. In the validation experiments, we inoculated each treatment in five or six replicate boxes with four plants each (p. 15 l. 9-21).

16. P11-Line 6: Please provide the full name of “At-LSPHERE”.

We added to the text an explanation about the origin of this name, which is not an acronym (p. 2 l. 31).

17. P12-Line 21: How to screen the Mini5SynComs? Please added references or related details.

We now transferred important information on the experimental procedures, which were initially summarized in “results” to the methods section in order to make the description of methods more self-contained and easier to read. The information on the screening of Mini5SynComs can be found in the method section: “SynCom size and strain pool size and composition for Mini5SynCom screen”, and “Synthetic community assembly and controls of the Mini5SynCom screen”.

18. P14-Line 1: Did perform normal distribution and homogeneity analysis of all data before using ANOVA and t-test analyses?

We knew from previous experience that the luminescence data are normally distributed after log₁₀-transformation. However, in response to the comment, we added the results of Shapiro-Wilk tests in Table S1. We confirmed normal distribution, as we could not reject the null hypothesis of our data being sampled from a normal distribution for the treatments use in ANOVA and t-test analyses.

19. P16-Line 6: Please remove the misadded brackets.

The brackets were meant to indicate the R function `varImp()`. We agree, it might be confusing, and we removed the brackets.

20. Figures: The first letter of the Y-axis and X-axis label is capitalized. Please revised throughout this manuscript.

We capitalized the first letter of the Y and X axis throughout.

Reviewer #2 (Remarks to the Author):

The authors demonstrate an approach to identify individual bacteria that robustly affect plant health in a community context. Specifically, they used random synthetic communities of bacteria to evaluate protection against a bacterial leaf pathogen, then used machine learning to identify strains that were strongly associated with protection against the pathogen. Most impressive was their very thorough validation, including 68 independent syncoms to validate the machine learning, thorough experiments to validate the protective strains, as well as experiments identifying other protective strains, which further showed that even slightly less “predictive” strains identified with machine learning were protective in combination. I think this is a novel approach until now, I really enjoyed the study and think that it could inspire a lot of interesting work in the future.

We thank the reviewer for the summary and the positive assessment.

A couple of comments are minor but are for me most important:

Based on Figure 8a. I could be wrong, but it seems like if one just had identified the clear bimodal distribution, then manually looked at what strains were most frequent in the “protective” syncoms that were not in the non-protective ones, one would likely find the same strains without complicated machine learning approaches that require special knowledge and extensive validation. Despite that, I think this is an important demonstration of an approach that can bring clear advantages. The authors could more clearly discuss in what sorts of screens this approach would really outperform manual analysis, i.e., when would manual analysis become unrealistic – with large pools, more complex communities, or screening for combinations of taxa or more complex traits, maybe even multiple traits perhaps?

Indeed, manual analyses might also have led to the identification of the protective candidates being more frequent in protective SynComs. However, statistical analyses offer the benefit of calculating the probability of differences observed in experiments when strains actually have the same frequencies in protective and non-protective SynComs. It offers a better control of the rates of type I errors (false positive). Regarding alternative statistical approaches to machine learning, differential-prevalence analyses could have been used. However, chi², Fisher’s exact tests, simpler decision trees and regression analyses do not implement procedures to avoid overfitting, which machine learning does. As a result, we would expect to identify more false protective strains with these alternative technics, leading to unsuccessful empirical validations (and thus a waste of resources due to the inflation of experimental conditions to test). In addition, the selection of feature importance implemented in the random forest analyses allows for identification of strains being important for protection despite an equal or even lower frequency in the “protected class”. We have now included a section in the discussion to compare the approaches (p. 10 l. 5-12): “When data were split among two classes, “protected” and “non-protected”, a simpler analytical approach might have consisted of identifying strains being more frequent in the

“protected” [...] random forest analyses enables the detection of key strains for plant protection across all possible patterns of differential frequencies within protection classes.”

The random forest and GLMNet methods implemented in the present study also offer large flexibility in their respective statistical frameworks. They can treat the response variable as categorical, discrete, or continuous. Continuous and discrete treatments avoid a-priory definition of classes which might often be difficult to justify (*e.g.*, unimodal distributions). In the case of the present study, the congruent results with the treatments of the pathogen colonization as a categorical or continuous variable reinforced the importance of candidate strains before their empirical validation as protective strains. These machine learning technics can also combine complex categorical, discrete, and continuous variables in one analysis to investigate the strength of their respective relationships with the response variable in an integrated context. We added this aspect to our discussion (p. 10 l. 1-5): “Random-forest and GLMNet approaches offer large flexibility [...] avoid the need for a-priory definition of classes, which, on the contrary to the present case, can often be difficult to justify in the absence of a bimodal distribution.”

Another point that I think might be important is that learning models are very common for predictions in complex systems and this is an adaptation of these approaches. I think some literature and discussion of this is missing. This includes fairly common use of for example random forest models in plant microbiome studies, mostly to identify taxa linked to specific plant organs, sampling sites, etc. The work here is clearly a distinct application from that, but still such studies could be cited in the discussion of what makes this work unique.

We agree. We now provide more context for our work and included examples where machine learning technics were applied (p. 9 l. 34-36): “Previous research has successfully utilized machine learning algorithms to identify individual microbial taxa associated with specific properties, including ecosystem functions, disease-suppressive soil, plant organs, and agricultural management^{7,58,63-66}.”

Further minor comments:

Intro

L9: “..., and ... - or...”

We modified the sentence to improve clarity.

Results

First section:

I think this section is a very nice overview of the considerations that go into the study design, which is very nice. However, in my opinion it distracts from the rest of the story, as other syncom sizes, pool sizes, etc. are ultimately not used. Thus in my opinion it could be supplemental information.

We shortened the text and added clarification in the methods section.

P3, L12: It struck me that the infection was allowed to go so long, which is unusual. I see in the methods that this was a very low inoculation dose, so that makes sense. Maybe it could be mentioned here that this was a low-dose infection to avoid confusion.

We adjusted the sentence as suggested.

In figure 2 and P3, L13-15, SynCom size is clearly defined as number of strains but Strain pool size is not really clearly defined. I thought first total number of cells based on Fig 2a, but this would not affect strain frequency, so it must be something else. Note: after reading on I realize the pool is referring to the pool of

strains drawn from (Fig 2b) and the frequency is how often the strains show up in communities. But this was not clear to me until later.

We clarified these definitions earlier in the text to facilitate reading.

P3, L35: I don't really have a problem with using CFUs (see also comments below), but I have trouble following the reasoning here. Are you essentially saying that luminescence is too insensitive? That is important, so perhaps that could be more explicit.

This reviewer is correct. The luminescence is less sensitive compared to pathogen numbers, especially when pathogen colonization is low. To increase sensitivity and the range of the pathogen read-out, we used CFU counting. We adjusted the text to improve clarity (p. 4 l. 8-11).

P4, L20: Is there a reason that the strain frequency plot in fig 2a has no axis scales?

Thank you for noting. We added the axes to the figure.

I think it would be nice for readers throughout the manuscript if Fig2b already had some more detailed taxonomy, like genus names.

We agree that it will provide the reader with valuable information. We modified Figure 2b to include the genus of the strains. More information on the taxonomy of the strains are documented in Supplemental Table 2.

Second section:

Methods related to pathogen CFU counting and Fig 3a: In our experience, on R2A many bacteria are look very similar and Rif-resistance occurs among leaf-colonizing bacteria. Do the authors have any assurance that DC3000 was not at least occasionally over-counted because of the other strains in the mix?

This is a valid concern, but in our system, we do not have the issue of over-counting the pathogen. We only have one strain in the At-LSPHERE that is resistant to Rif (*Aeromicrobium* Leaf289), and it grows more slowly and has a different morphology than the pathogen (DC3000). We see the value of including this information and added an explanation in the methods section (p. 12 l. 29-32).

Methods related to commensal CFU counting and Fig3b: It is not completely clear to me - the "commensal" counts were done on non-selective media, so do they include DC3000, or was it always excluded or its counts on selective media subtracted from totals? Perhaps that could be mentioned more clearly in the results/methods.

We can distinguish the pathogen from the commensal colonies and can count them separately. We added an explanation of how we counted the individual strains in the methods section (p. 12 l. 31 – p. 13 l. 31).

Fig 3: The figure header says there is a scatterplot overlaid, but I cannot see it.

We apologize, it seems that during the export to pdf, the scatterplot was lost. It is now included.

Third section:

P5, L19-20: If the pathogen is included in the commensal levels (see previous comment) would that explain this positive correlation?

The pathogen is not included in the commensal count. We have clarified this now (see response above).

Fig 4: In the header, “Fixed effect in the model” – maybe referring to the details in the methods section would help the figure/heading stand alone better.

The figure header was adjusted accordingly.

Fourth section:

I realize the methods of this part are very important for the results, but it should be noted that this section reads a lot like methods. I’m not really sure how much this can be improved upon, but I think some details could be relegated to the methods section.

We transferred experimental information to the Methods section, as suggested.

The verification of the machine learning models is really nicely done with an independent data set. However, it was not clear to me if the authors checked how many of the 68 new Mini5SynComs were already represented in the original set – I would think that it would be important for them to be new combinations to really check the predictions.

The reviewer is right, and we had made sure that none of the 68 newly assembled Mini5SynComs in the test dataset overlapped with the training data set. We have clarified this in the text (p. 6 l. 20-22, p. 18 l. 23-29).

Fig 5: The meaning of “presence/absence” and “colonization” groups are not clear from the figure heading. RF and GLMNet are not in the legend either.

We revised the figure heading and legend.

P6, L17: Are abundances referring to counts at the end of the experiment, I guess?

This is correct. The abundances are referring to CFU per gram fresh weight at the end of the experiment. We clarified the corresponding text p. 18 l. 9-10: “Predictors were either presence/absence of Mini5SynCom members or the absolute abundance of these determined by CFU enumeration at the end of the experiment (“colonization”).”

P6, L29-32: I am not an expert on this, but I think precision is true positives / (true positives + false positives) and recall is true positives / (true positives + false negatives). I think the way the definitions are written here would be identical or the difference hasn’t been made clear.

The reviewer correctly defines the two metrics. We rephrased the corresponding text to make the difference clearer. It now reads “The fraction of true protective samples in the set of samples predicted as protective (*i.e.*, precision) [...]. The fraction of correctly predicted protective samples in the set of true protective samples (*i.e.*, recall) [...]” (p. 6 l. 27-31).

Fifth section:

The use of random strains and combinations is a really nice control for this experiment, well done.

Thank you.

P7, L33: This is true, but also the other pairs seem to be comparable.

We added a clarification (p. 8 l. 4-7): “Other binary PR-strain combinations showed non-significant lower

pathogen colonization compared to their individual treatments despite a significant reduction of pathogen colonization compared to axenic controls (Fig. 7a, Supplementary Table 11).” The binary combination of Leaf68-Leaf76 is the only binary combination to significantly reduce pathogen colonization below level of individual strains, the other two are intermediate and non-significant to the pathogen colonization with individual strains.

Sixth section:

Fig8: ValMix and Mix are not clearly explained in the figure heading

We modified the figure caption to clarify. ValMix is an abbreviation for validation mix, which is a new Mini5SynCom, while Mix6 is a repetition from the screen to prove reproducibility. We clarified in the methods section as well (p. 15 l. 26-32).

Discussion:

The authors do a very nice job of discussing the results. Throughout, I was wondering if machine learning could be implemented to identify combinations, besides the manual identification used here. The authors do touch on this in L28-31. Maybe they could go a bit deeper, for example I note that in the PR Strains mixes (Fig 8a), there is a very large variation over 4 orders of magnitude – can this be used further to identify “helper” strains or others that negatively affect the effectiveness of a useful strain, besides just combinations of the most predictive ones?

Indeed, with a higher prevalence of strains in the Mini5SynCom than in the present study, the structure of models used in GLMNet, and the features and permutation procedures of feature importance in Random Forest could be set to specifically investigate the presence of “helpers” or “cancellers” of protective strains. We expanded on this point in the manuscript as suggested (p. 10 l. 12-26).

Regarding variation in the data from our screen. Our validation experiments (Figure 7a) showed that combinations of PR strains reduce pathogen colonization further than when they were present by themselves. This is in-line with variation of pathogen colonization in the “with PR strains” group in Figure 8a. The Mini5SynComs of that group contain either one, two or all three of the PR strains. We now clarified this point in the manuscript (p. 8 l. 18-20).

Looking at Supp Fig2c, most results agreed with Vogel et al., but some not (e.g., Acidovorax 76 was not protective there and Strain 21 was not very predictive of protection here). Maybe the authors could briefly discuss what might cause the differences.

This is correct. We now mention these differences. Note that the experimental system was different which might explain these incongruences. Here, we grew the plants in large microboxes with a calcined-clay substrate, while the individual screen of 224 strains in Vogel et al. was conducted in 24 well plates with agar substrate.

Reviewer #3 (Remarks to the Author):

The authors aimed to identify the beneficial microbiome that can protect plant health. They used experimental and machine learning methods to randomize and simplify the synthetic communities (SynComs), and then examined the disease resistance of these SynComs. The study provides strong evidence for the ability of SynComs to enhance host adaptation. Additionally, the framework proposed in this paper is important for efficiently constructing synthetic communities in future studies. However, there are a few issues that should be addressed. Notably, the absence of high-throughput sequencing data, plant

resistance gene response data, and quantitative data from pathogen qPCR analysis in this paper is significant. Including these components is crucial for reaching a more reliable conclusion. Due to these limitations, I have significant reservations about the findings of this paper. Please refer to my detailed review comments below.

1.The introduction lacks coherence and fails to provide a comprehensive overview of the development of the synthetic microbiome and its current challenges. Furthermore, it does not clearly state a well-defined scientific question.

Thank you for this comment. We have revisited and rewritten the introduction to better introduce current challenges and explicitly state the goal of the study: “In this study, we present an experimental and analytical approach to address the question of which microbiota features support plant protection across different biotic contexts.” We now also highlight the general and ongoing interests in ecology to link our study to open questions in microbial ecology.

2.The manuscript would benefit from including amplicon data to track the dynamics of microbial communities. It is also important to include negative controls and conduct amplicon sequencing to determine if the SynComs were affected by contamination from environmental bacteria and fungal spores.

Amplicon sequencing is indeed an alternative approach to investigate relative abundance of strains, and we have applied it in a previous screen where we wanted to differentiate 15 strains (ref 76). However, in the present study we chose to quantify community composition by direct counts. The method allows the generation of absolute cell numbers and to avoid biases such as cell lysis efficiency, primer binding preferences and 16S rRNA copy number. Studies using time series analyses to investigate the dynamics of communities are valuable, but the present study aims to introduce an efficient approach which allows robust and large screens to identify causes of microbiota-conferred phenotype. In this context, a simple read out from a unique time point is considered an advantage regarding the ratio of resource consumption to the number of situations screened. The sufficiency of this “single point” read out was confirmed by our experimental validations.

Regarding contamination, our experiments are performed under gnotobiotic conditions to avoid contamination from environmental bacteria and fungal spores. The use of CFU has the added advantage that we can control potential contamination, as we would be able to detect any potential contamination (based on their (known) colony colour, morphology and shape). As control conditions, we also included axenic plants to which we applied mock inoculation and infection, and we did not detect any contamination.

3.The author should provide detailed information regarding the machine learning analysis, including the missing error rates and Tenfold cross-validation curves.

We added more details regarding machine learning analysis in the method section. However, we are not sure to understand what the reviewer meant by “the missing error rates”. To our knowledge, this term is not common in the field of machine learning. Metrics of performances are presented in Tables S6 and S7. For classification analyses, all frequencies of true and false results are presented along with Accuracy, Kappa, Specificity, Precision, Recall, F1, and Balanced accuracy. For the regression analyses, we report the root mean square errors.

Validation curves consisting in plotting performances across values of an hyperparameter for training and validation sets is sometimes used to estimate overfitting sensitivity to the values of an hyperparameter. However, the tuning of the models is based on the maximization (or minimization) of the scoring function on the validation sets. Consequently, the only real estimation of the generalization of predictions relies of the evaluation of performances on an adequate independent test set. In the present study, we generated an independent test set from a new experiment (Experiment 3), and we made sure that none of the communities used were seen by the models during training. For this reason, we feel that adding validation

curves will add unnecessary complexity to the results and might mislead readers not familiar with machine learning techniques.

4.Extracting a cohesive conclusion from this manuscript, which incorporates all the presented data, is challenging. This difficulty primarily arises from shortcomings in the experimental design, such as the absence of microbial amplicon data, the lack of a negative control in unplanted soil samples, and the omission of plant-resistant gene/pathway-related data.

With the approach used in the present study, we provide absolute commensal colonization data, and we believe that microbial amplicon data would not provide better information regarding abundance data (see response above). Regarding unplanted soil, we worked under gnotobiotic conditions with plants grown enclosed in microboxes with non-natural sterile substrate (ref 53, 55, 76, 79). We included negative controls (mock-inoculated and mock-infected plants, referred to as axenic) and thoroughly validated axenic conditions for both plants and substrate and did not detect any contaminations. We now added more details about the substrate used, its sterilization, the growing conditions, and control used, to improve clarity (p. 11 l. 23-32, p. 12 l. 9-18, p. 13 l. 28-31, p. 14 l. 29 – p. 15 l. 1).

We agree that the use of plant-resistant genes and pathway-related data might give insights into protection mechanisms; however, in addition also resource competition and other mechanisms might be involved. Identifying the mechanisms of protection, however, was not the goal of this study. We hope the adjustments made in the introduction clarify this point.

5.The evidence or data supporting the selection of the five bacteria for the SynCom is not sufficient, although it is mentioned that they were obtained from a previous pool of 137 strains.

We describe the process now in more detail in the Methods section (p13 l. 24-26, p. 14 l. 24-27). We did not select the five strains of each community following specific criteria, but instead randomly assembled those five-strain communities from a pool of 35 strains.

6.The excessive use of the active voice has negatively impacted the readability of this article.

We reduced the use of active voice.

7.Relying solely on the dilution smear plate method for quantitative analysis of synthetic communities and pathogenic bacteria poses uncertainties, as this method is relatively unreliable and subjective. I strongly recommend incorporating second or third-generation high-throughput sequencing for quantitative analysis and dynamic tracking, which will provide more reliable verification of abundance changes in the SynComs and enable confirmation of whether the selected three bacteria are indeed the core bacteria.

We are happy to clarify, we did not streak out cell suspensions. We used a previously optimized protocol of spotting and plating of dilution series, which is highly reliable (references 22, 45, 76, 79). The correlation between CFU counting and i.e. luminescence of the pathogen has also been demonstrated (references 20, 22, 45, 80). We prefer CFU determination over amplicon sequencing to generate absolute bacterial counts and to avoid biases in cell lysis, primer binding, copy number that make CFU counting more accurate (see our reply above). We clarified this technical aspect in the methods section (p. 12 l.29 – p. 13 l.21).

8.It is necessary for the author to include qPCR and RNA-seq data from plants to substantiate the disease-resistant ability of the synthetic communities (SynComs). It is also crucial to perform absolute quantification of pathogenic bacteria using qPCR.

This must be a misunderstanding; we assessed pathogen reduction, which is a different trait of a plant-pathogen systems. For this reason, the generation of RNA-seq data or qPCR of plant defence markers

would not add to the objectives of our study. We determined absolute number of the pathogen from a reliable method using spotting and dilution series (see response above). The advantage of colonization data over qPCR for pathogen prevalence, other than having absolute colonization, is that we account for live pathogen only, which gives us indication for how fit and virulent the pathogen remains *in planta*.

9. There is a lack of useful information in the major Figures, such as Figure 1, Figure 2, Figure 3, and Figure 4.

Figure 1 is a demonstration of the experimental study design. We consider it important to provide an overview of the study design. Figure 2 includes the strain pool of the screen. Figure 3 shows the results of the Mini5SynCom screen, and Figure 4 the evaluation of the machine learning methods applied. We feel that all figures are valuable for the study presented.

10. Furthermore, there is no explanation of the pathogenicity index or provision of images related to diseased plants.

This seems to be a misunderstanding; we do not provide a pathogenicity index. Instead, we use pathogen colonization as our main readout as it is quantitative and continuous. We have previously shown that pathogen number and disease correlate (ref 20, 22, 45). For documentation and transparency reasons, we agree that providing the original images of plants is necessary and invaluable, and we now provide them (<https://doi.org/10.5281/zenodo.8399345>), as well as reference to the images in the methods section: "Plant images at 14 dpi are available through Zenodo."

11. Furthermore, exploring genes involved in plant-phyto microbiome signaling pathways, such as those responsible for LCOs and bacteriocins, and investigating their response to pathogens would be valuable.

The identification of the protection mechanisms of the strains is outside of the scope of this study. This study aimed to demonstrate the use of machine learning methods and a screen for beneficial individual and combination of strains in a community context. Only exploring genes of signalling pathways or antibiotic compounds also omits the various other ways that beneficial microbes can counteract the pathogen, i.e. resource competition, virulence suppression. However, we agree, this will be interesting to explore in future work.

12. In the "Strain pool size for community screen" section of Methods, why should each strain appear in 20 communities and not more? In addition, $E(X) = [k!(n-1)! / n!(k-1)!] * N$, isn't that equal to $(k/n) * N$?

Regarding statistical results, the higher is the strain prevalence the higher the predictive power becomes. However, this prevalence depends on the number of plants which can be handled experimentally, the size of SynComs, and the number of strains being screened (strain pool size). Because we could include a maximum of 544 plants in the screening experiment (distributed across 136 microboxes) and we decided to use five-strains communities, the remaining tuneable parameter was the strain pool size. We estimated that an average prevalence of 20 was reasonable to increase the chance of achieving robust statistical results. We adjusted the Methods section to centralize all methodological aspects in this part of the manuscript and hope that it clarifies this point (p. 13 l. 23 – p. 14 l. 13).

The reviewer is right, about further simplification of the equation. It has been adjusted (p. 14 l. 8).

13. The paragraph 3 of introduction, which introduces the *Arabidopsis thaliana* phyllosphere is good material for exploring these questions. But it is not coherent to move directly to machine learning. The background on machine learning for mining beneficial microorganisms is not well explained. The key questions addressed by the study are not well expressed.

We reworked the introduction to better highlight the rationale for ML usage and the addressed questions.

14. Whether the synthetic community developed in this study to identify beneficial microorganisms is applicable to other plants? This could be discussed.

We agree that this is an interesting discussion point and added it (p. 10 l. 37 – p. 11 l. 2).

15. The significance test symbol (abcd) in Figure 8d may be incorrect? and the "Bonf. corrected p-value" in Table S11, S12 and S15 should not be greater than 1.

Thank you for noting. The reviewer is correct, one significance symbol was missing in Fig. 8d, because the code to add the symbols did not include this level. This is now corrected. Regarding the probability of a p-value. As a probability it cannot be greater than one, this is correct. However, in the present study we employed the approach consisting in multiplication of each p-value by the number of comparisons to obtain a Bonferroni corrected p-values rather than correcting the alpha level. This explains why some corrected p-values are greater than one in the mentioned Supplemental Tables. A common practice consists of arbitrarily setting all Bonferroni-corrected p-values above one to be one. We now follow this practice in the manuscript.

16. Reference 13 not following journal style?

Corrected.

REVIEWERS' COMMENTS

Reviewer #1 (Remarks to the Author):

The authors have improved the description of the methods and results sections, addressing the issues raised previously. However, I share Reviewer-3's concerns, although the focus of this article is on demonstrating the use of machine learning methods and a screen for beneficial individuals and a combination of strains in a community context, the most direct and effective indicator to evaluate the effectiveness of this method is the strength of the plant immune response after application of the beneficial strain. Based on the biomass of pathogen colonization alone, plant disease resistance and immune response cannot be fully reflected. I strongly recommend that the authors provide information related to plant immune response or interaction between plant and pathogen to verify the effectiveness of this method and highlight its application value in the future.

Reviewer #2 (Remarks to the Author):

I have carefully reviewed the responses of the authors and the changes in the manuscript. The authors have done an excellent job addressing my comments and I think they have also addressed most of the other comments, as far as I can tell. The manuscript is in very good shape. I do not have further comments.

Reviewer #3 (Remarks to the Author):

The author has taken into account all my comments and responded to them in detail, either with necessary corrections or with their rebuttal/explanation on several points. I agree with their explanation and clarification of the aim of this study. Although the paper is still limited by the lack of plant resistance genes and pathway-related data to provide insight into the protective mechanisms of the syncom (i.e. whether a response of the plant immune system or competition among microorganisms), the paper does provide a useful protocol to provide a framework that can be adapted to identify features relevant to microbiota function.

REVIEWERS' COMMENTS

Reviewer #1 (Remarks to the Author):

The authors have improved the description of the methods and results sections, addressing the issues raised previously. However, I share Reviewer-3's concerns, although the focus of this article is on demonstrating the use of machine learning methods and a screen for beneficial individuals and a combination of strains in a community context, the most direct and effective indicator to evaluate the effectiveness of this method is the strength of the plant immune response after application of the beneficial strain. Based on the biomass of pathogen colonization alone, plant disease resistance and immune response cannot be fully reflected. I strongly recommend that the authors provide information related to plant immune response or interaction between plant and pathogen to verify the effectiveness of this method and highlight its application value in the future.

Thank you for the comment.

As the reviewer states, the purpose of our study was to demonstrate the use of machine learning methods to identify community features in a community context. We are pleased to learn that the reviewer finds that the previous issues are now addressed.

Regarding the protection mechanisms, we agree that knowledge of the protection mechanisms will be important, and we have included a discussion on protection mechanisms of beneficial strains in our manuscript, page 10 l. 27-35. Apart from the activation of plant defences, protective bacteria can reduce pathogen growth by resource competition, or by direct killing (e.g. PMID: 34819644). Previous work (e.g. PMID: 34007033) has shown that the strength of the plant response to commensal bacteria does not necessarily correlate with the commensal's protection potential. We therefore argue that a mechanistic exploration of the disease resistance should involve an integrative exploration of the relative importance of both plant response and bacterial interactions. Our approach, using pathogen colonization as a readout, captures the potential combined effects of all these mechanisms.

Reviewer #2 (Remarks to the Author):

I have carefully reviewed the responses of the authors and the changes in the manuscript. The authors have done an excellent job addressing my comments and I think they have also addressed most of the other comments, as far as I can tell. The manuscript is in very good shape. I do not have further comments.

Thank you for your kind feedback.

Reviewer #3 (Remarks to the Author):

The author has taken into account all my comments and responded to them in detail, either with necessary corrections or with their rebuttal/explanation on several points. I agree with their explanation and clarification of the aim of this study. Although the paper is still limited by the lack of plant resistance genes and pathway-related data to provide insight into the protective mechanisms of the syncom (i.e. whether a response of the plant immune system or competition among microorganisms), the paper does

provide a useful protocol to provide a framework that can be adapted to identify features relevant to microbiota function.

Thank you for your kind feedback.